# THE CONDITIONAL ENTROPY BOTTLENECK

## ABSTRACT

We present a new family of objective functions, which we term the *Conditional Entropy Bottleneck* (CEB). These objectives are motivated by the *Minimum Necessary Information* (MNI) criterion. We demonstrate the application of CEB to classification tasks. We show that CEB gives: well-calibrated predictions; strong detection of challenging out-of-distribution examples and powerful whitebox adversarial examples; and substantial robustness to those adversaries. Finally, we report that CEB fails to learn from *information-free* datasets, providing a possible resolution to the problem of generalization observed in Zhang et al. (2016).

## 1 INTRODUCTION

The field of Machine Learning has suffered from the following well-known problems in recent years[1]:

- **Vulnerability to adversarial examples.** Essentially all machine-learned systems are currently believed by default to be highly vulnerable to adversarial examples. Many defenses have been proposed, but very few have demonstrated robustness against a powerful, general-purpose adversary. Lacking a clear theoretical framework for adversarial attacks, most proposed defenses are ad-hoc and fail in the presence of a concerted attacker (Carlini & Wagner, 2017a; Athalye et al., 2018).

- **Poor out-of-distribution detection.** Classifiers do a poor job of signaling that they have received data that is substantially different from the data they were trained on. Ideally, a trained classifier would give less confident predictions for data that was far from the training distribution (as well as for adversarial examples). Barring that, there would be a clear, principled statistic that could be extracted from the model to tell whether the model *should* have made a low-confidence prediction. Many different approaches to providing such a statistic have been proposed (Guo et al., 2017; Lakshminarayanan et al., 2016; Hendrycks & Gimpel, 2016; Liang et al., 2017; Lee et al., 2017; DeVries & Taylor, 2018), but most seem to do poorly on what humans intuitively view as obviously different data.

- **Miscalibrated predictions.** Related to the issues above, classifiers tend to be very overconfident in their predictions (Guo et al., 2017). This may be a symptom, rather than a cause, but miscalibration does not give practitioners confidence in their models.

- **Overfitting to the training data.** Zhang et al. (2016) demonstrated that classifiers can memorize fixed random labelings of training data, which means that it is possible to learn a classifier with perfect *inability* to generalize. This critical observation makes it clear that a fundamental test of generalization is that the model should *fail* to learn when given what we call *information-free* datasets.

This paper does not set out to solve any of these problems. Instead, our sole interest is the learning of optimal representations. In pursuit of that goal, we attempt to be as general as possible, considering only how to define optimal representations, what objective function might be capable of learning them, and what requirements such an objective function places on the form of the model.

Given an optimal (according to our criterion) objective function, however, it is natural to explore the problems listed above, to see if such an objective function can ameliorate some of the core issues in the field of machine learning. We make those explorations in this paper, and find that our objective function, the *Conditional Entropy Bottleneck* (CEB) appears to impact all of the issues listed above.

---

[1]These problems existed before recent years, but not all of them were known. In particular, adversarial examples were unknown prior to 2013, and the severity of the overfitting problem was not known until 2016.

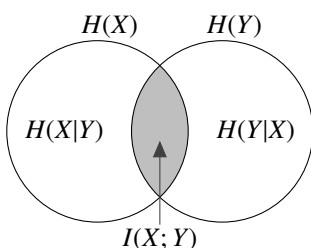 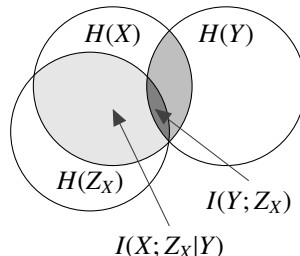

Figure 1: **(Left)**: Information Venn diagram showing the joint distribution over $X, Y$. **(Right)**: The joint distribution $Z_X \leftarrow X \leftrightarrow Y$. $Z_X$ is carefully positioned to indicate its conditional independence from $Y$ given $X$.

## 2   OPTIMAL REPRESENTATIONS

Consider a joint distribution, $p(x, y)$, represented by the graphical model:

$$X \leftrightarrow Y$$

This joint distribution is our data, and may take any form. We don't presume to know how the data factors. It may factor as $p(x, y) = p(x)p(y|x)$, $p(x, y) = p(y)p(x|y)$, or even $p(x, y) = p(x)p(y)$.

The first two factorings are depicted in Figure 1 in a standard information diagram showing the various entropies and the mutual information. We can ask: given this generic setting, what is the optimal representation? It seems there are only two options: capture all of the information in both $X$ and $Y$ (measured by the joint entropy, $H(X, Y)$), or capture only the information shared between $X$ and $Y$ (measured by the mutual information, $I(X; Y)$).

The field of lossless compression is concerned with representations that perfectly maintain all of the information in both $X$ and $Y$, as are the closely related studies of *Kolmogorov Complexity* (Kolmogorov, 1965) and *Minimum Description Length* (MDL) (Grünwald, 2007), all three of which are concerned with perfect *reconstruction* of inputs or messages.

In contrast, we think that the field of machine learning is primarily concerned with making optimal *predictions* on unseen data. The requirements of perfect reconstruction from a compressed representation may result in the retention of much more information in the model than may be needed for prediction or stochastic generation tasks. For most such machine learning tasks, this points towards learning representations that capture only the information shared between $X$ and $Y$, which is measured by the mutual information, $I(X; Y)$.

The mutual information is defined in a variety of ways; we will use two (Cover & Thomas, 2006):

$$I(X; Y) = H(X) - H(X|Y) = H(Y) - H(Y|X) \tag{1}$$

$I(X; Y)$ measures the amount of information necessary to define the relationship between $X$ and $Y$. For some fixed dataset $X, Y$, any information less than $I(X; Y)$ must be insufficient to predict $Y$ from $X$ or vice-versa with minimal error. Equivalently, any information more than $I(X; Y)$ must contain some superfluous information for those two tasks. For example, consider a labeled dataset, where $X$ is high-dimensional and information-rich, and $Y$ is a single integer. All of the information in $X$ that is not needed to correctly predict the single value $Y = y$ is useless for the prediction task defined by the dataset, and may be harmful to the performance of a machine learning system if retained in the learned representation, as we will show empirically below. Next, we formalize this intuition about the information required for an optimal representation.

## 3   MINIMUM NECESSARY INFORMATION

We propose the *Minimum Necessary Information* (MNI) criterion for a learned representation. We can define MNI in three parts. First is *Information*: we would like a representation that captures semantically meaningful information. In order to measure how successfully we capture meaningful

information, we must first know how to measure information. Thus, the criterion prefers information-theoretic approaches, given the uniqueness of entropy as a measure of information (Shannon, 1948). The semantic value of information is given by a task, which is specified by the set of variables in the dataset. I.e., the dataset $X, Y$ defines two tasks: predict $Y$ given $X$, or predict $X$ given $Y$. This brings us to *Necessity*: the information we capture in our representations must be necessary to solve the task.[2] Finally, *Minimality*: this simply refers to the amount of information – given that we learn a representation that can solve the task, we require that the representation we learn retain the smallest amount of information about the task out of the set of all representations that solve the task. This part of the criterion restricts us from incorporating "non-semantic" information into our representation, such as noise or spurious correlation.

More formally, in the case of two observed variables, $X$ and $Y$, a necessary set of conditions for a representation $Z$ to satisfy the MNI criterion is the following:

$$I(X; Y) = I(X; Z) = I(Y; Z) \tag{2}$$

This fully constrains the *amount* of information.

To constrain the *necessity* of the information in the representation $Z$, the following conditions must be satisfied:

$$p(y|x) = \int dz\, p(y|z)p(z|x) \qquad p(x|y) = \int dz\, p(x|z)p(z|y) \tag{3}$$

These four distributions of $z$ correspond to the two tasks: predict $Y$ given $X$ and predict $X$ given $Y$.[3]

## 4 THE CONDITIONAL ENTROPY BOTTLENECK

One way to satisfy Equation (2) is to learn a representation $Z_X$ of $X$ only, indicated by the Markov chain $Z_X \leftarrow X \leftrightarrow Y$. We show this Markov chain as an information diagram in Figure 1 (Right). The placement of $H(Z_X)$ in that diagram carefully maintains the conditional independence between $Y$ and $Z_X$ given $X$, but is otherwise fully general. Some of the entropy of $Z_X$ is unassociated with any other variable; some is only associated with $X$, and some is associated with $X$ and $Y$ together. Figure 1 (Right), then, shows diagrammatically the state of the learned representation early in training. At the end of training, we would like $Z_X$ to satisfy the equalities in Equation (2), which corresponds to Figure 1 (Left), where the gray region labeled $I(X; Y)$ also corresponds to $I(X; Z_X)$ and $I(Y; Z_X)$.

Given the conditional independence $Z_X \perp\!\!\!\perp Y|X$ in our Markov chain, $I(Y; Z_X)$ is maximal at $I(X; Y)$, by the data processing inequality. However, $I(X; Z_X)$ does not clearly have a constraint that targets $I(X; Y)$. We cannot maximize $I(X; Z_X)$ in general while being compatible with the MNI criterion, as that is only constrained from above by $H(X) \geq I(X; Y)$. Instead, we could use the Information Bottleneck objective (Tishby et al., 2000) which starts from the same Markov chain and minimizes $\beta I(X; Z_X) - I(Y; Z_X)$, but it is not immediately clear what value of $\beta$ will achieve the MNI.

Thus, we need a different approach to hit the MNI. Considering the information diagram in Figure 1 (Left), we can notice the following identities when when we have achieved the MNI:

$$I(X; Y|Z_X) = I(X; Z_X|Y) = I(Y; Z_X|X) = 0 \tag{4}$$

With our Markov chain and the chain rule of mutual information (Cover & Thomas, 2006), we have:

$$I(X; Z_X|Y) = I(X, Y; Z_X) - I(Y; Z_X) = I(X; Z_X) - I(Y; Z_X) \tag{5}$$

This conditional information is guaranteed to be non-negative, as both terms are mutual informations, and the Markov chain guarantees that $I(Y; Z_X)$ is no larger than $I(X; Z_X)$, by the data processing inequality. From an optimization perspective, this is ideal – we have a term that we can minimize, and we can directly know how far we are from the optimal value of 0 (measured in nats, so it is

---

[2]Achille & Soatto (2018) and other authors call this "sufficiency". We avoid the term because it leads to confusion with minimum sufficient statistics, which maintain the mutual information between data and the true model that generates it.

[3]Note that a $\delta$ distribution for $p(z|x)$ and $p(z|y)$ could satisfy this condition, but those distributions would not satisfy Equation (2) unless $Y = f(X)$ and $X = g(Y)$. Anything other than a bijective relationship would give $I(X; Y) < \max(H(X), H(Y))$.

interpretable), when we are done (when it's close enough to 0 that we are satisfied), and when our model is insufficient for the task (i.e., when this term *isn't* close enough to 0). This leads us to the general *Conditional Entropy Bottleneck* objective:

$$\text{CEB} \equiv I(X; Z_X|Y) - I(Y; Z_X) \tag{6}$$

Typically we would add a Lagrange multiplier on one of the two terms. In Appendix A, we present some geometric arguments to prefer leaving the two terms balanced.

It is straightforward to turn this into a variational objective function that we can minimize. Taking the terms in turn:[4]

$$I(X; Z_X|Y) = I(X; Z_X) - I(Y; Z_X) = H(Z_X) - H(Z_X|X) - H(Z_X) + H(Z_X|Y) \tag{7}$$

$$= -H(Z_X|X) + H(Z_X|Y) = \langle \log e(z_X|x) \rangle - \langle \log p(z_X|y) \rangle \tag{8}$$

$$\leq \langle \log e(z_X|x) \rangle - \langle \log p(z_X|y) \rangle + \text{KL}[p(z_X|y)\|b(z_X|y)] \tag{9}$$

$$= \langle \log e(z_X|x) \rangle - \langle \log b(z_X|y) \rangle \tag{10}$$

$e(z_X|x)$ is our *encoder*. It is not a variational approximation, even though it has learned parameters. $b(z_X|y)$ is the *backward encoder*, a variational approximation of $p(z_X|y)$.

In the second term, $H(Y)$ can be dropped because it is constant with respect to the model:

$$I(Y; Z_X) = H(Y) - H(Y|Z_X) \Rightarrow -H(Y|Z_X) = \langle \log p(y|z_X) \rangle \tag{11}$$

$$\geq \langle \log p(y|z_X) \rangle - \text{KL}[p(y|z_X)\|c(y|z_X)] \tag{12}$$

$$= \langle \log c(y|z_X) \rangle \tag{13}$$

$c(y|z_x)$ is the *classifier* (although that name is arbitrary, given that $Y$ may not be labels), which variationally approximates $p(y|z_X)$.

The variational bounds derived above give us a fully tractable objective function that works on large-scale problems and supports amortized inference, *Variational Conditional Entropy Bottleneck* (VCEB):

$$\text{CEB} \equiv I(X; Z_X|Y) - I(Y; Z_X) \Rightarrow \langle \log e(z_X|x) \rangle - \langle \log b(z_X|y) \rangle - \langle \log c(y|z_X) \rangle \equiv \text{VCEB} \tag{14}$$

The distributions with letters other than $p$ are assumed to have learned parameters, which we otherwise omit in the notation. In other words, all three of $e(\cdot)$, $b(\cdot)$, and $c(\cdot)$ have learned parameters, just as in the encoder and decoder of a normal VAE (Kingma & Welling, 2014), or the encoder, classifier, and marginal in a VIB model.

We will name the $I(X; Z_X|Y)$ term the *Residual Information* – this is the excess information in our representation beyond the information shared between $X$ and $Y$:

$$Re_{X/Y} \equiv \langle \log e(z_X|x) \rangle - \langle \log b(z_X|y) \rangle \geq -H(Z_X|X) + H(Z_X|Y) = I(X; Z_X|Y) \tag{15}$$

There are a number of natural variations on this objective. We describe a few of them in Appendix E.

## 5 THE INFORMATION BOTTLENECK

The Information Bottleneck (IB) (Tishby et al., 2000) learns a representation of $X$ and $Y$ subject to a soft information constraint:

$$IB \equiv \min \beta I(Z; X) - I(Z; Y) \tag{16}$$

where $\beta$ controls the size of the constraint.

In Figure 2 we show the optimal surfaces for CEB and IB, labeling the MNI point on both. In Figure 4 we show the same surfaces for finite models and that adjusting $\beta$ determines a unique point in these information planes relative to $I(X; Y)$.

As described in Tishby et al. (2000), IB is a tabular method, so it is not usable for amortized inference.[5] Two recent works have extended IB for amortized inference. Both of these approaches

---

[4] We write expectations $\langle \log e(z_X|x) \rangle$. They are always with respect to the joint distribution; here, that is $p(x, y, z_X) = p(x, y)e(z_X|x)$.

[5]The tabular optimization procedure used for IB trivially applies to CEB, just by setting $\beta = \frac{1}{2}$. A recent work on IB using tabular methods is the *Deterministic Information Bottleneck* Strouse & Schwab (2017), which learns hard clusterings, rather than the soft clusterings of earlier IB approaches.

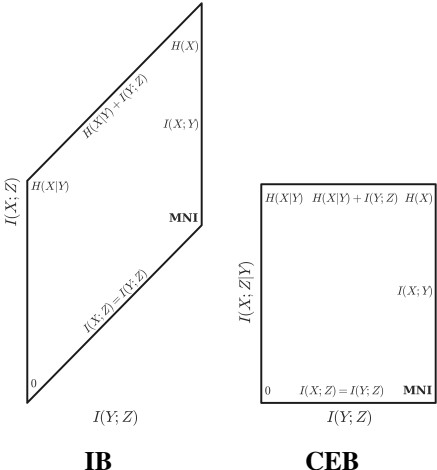

Figure 2: Geometry of the optimal surfaces for IB and CEB, with all points labeled. CEB rectifies IB's parallelogram by subtracting $I(Y;Z)$ at every point.

rely on sweeping $\beta$, and do not propose a way to set $\beta$ directly to train models where $I(X;Z) = I(Y;Z) = I(X;Y)$. Achille & Soatto (2018) presents *InfoDropout*, which uses IB to motivate a variation on Dropout (Srivastava et al., 2014). A varational version of IB is presented in Alemi et al. (2017). That objective is the *Variational Information Bottleneck* (VIB):

$$VIB \equiv \beta(\langle \log e(z_X|x)\rangle - \langle \log m(z_X)\rangle) - \langle \log c(y|z_X)\rangle \tag{17}$$

Instead of the backward encoder, VIB has a *marginal posterior*, $m(z_X)$, which is a variational approximation to $e(z_X) = \int dx\, p(x)e(z_X|x)$. Additionally, it has a hyperparameter, $\beta$. We show in Appendix A that the optimal value for $\beta = \frac{1}{2}$ when attempting to adhere to the MNI criterion.

Following Alemi et al. (2018), we define the *Rate* ($R$):

$$R \equiv \langle \log e(z_X|x)\rangle - \langle \log m(z_X)\rangle \geq I(X;Z_X) \tag{18}$$

We can compare variational CEB with VIB by taking their difference at $\beta = \frac{1}{2}$. Note that both objectives have an elided dependence on $\langle \log p(y)\rangle$ from the $I(Y;Z_X)$ term that we must track:

$$CEB - VIB_{\beta=\frac{1}{2}} = \langle \log b(z_X|y)\rangle - \langle \log m(z_X)\rangle - \langle \log c(y|z_X)\rangle + \langle \log p(y)\rangle \tag{19}$$

Solving for $m(z_X)$ when that difference is 0:

$$m(z_X) = \frac{b(z_X|y)p(y)}{c(y|z_X)} \tag{20}$$

Since the optimal $m^*(z_X)$ is the marginalization of $e(z_X|x)$, at convergence we must have:

$$m^*(z_X) = \int dx\, p(x)e(z_X|x) = \frac{p(z_X|y)p(y)}{p(y|z_X)} \tag{21}$$

Depending on the distributional families and the parameterizations, this point may be difficult to find, particularly given that $m(z_X)$ only gets information about $y$ indirectly through $e(z_X|x)$. Consequently, for otherwise equivalent models, we may expect $VIB_{\frac{1}{2}}$ to converge to a looser approximation of $I(X;Z) = I(Y;Z) = I(X;Y)$ than CEB. Since VIB optimizes an upper bound on $I(X;Z)$, that means that $VIB_{\frac{1}{2}}$ will report $R$ converging to $I(X;Y)$, but will capture less than the MNI. In contrast, if $Re_{X/Y}$ converges to 0, the variational tightness of $b(z_X|y)$ to the optimal $p(z_X|y)$ depends only on the tightness of $c(y|z_X)$ to the optimal $p(y|z_X)$.

Table 1: Accuracy and rates ($R$) for each model. **Bold** indicates the best score in that column. Determ doesn't have a rate, since it doesn't have an explicit encoder distribution. The final rate for the other four models is reported, as well as the peak rate achieved during training. The true mutual information for Fashion MNIST is $I(X; Y) = 2.3$ nats, so achieving $R = 2.3$ is optimal according to MNI.

| Model | Accuracy | Train $R$ final (peak) |
|---|---|---|
| Determ | **92.7** | n/a |
| $VIB_{0.01}$ | **93.0** | 2.6 (11.6) |
| $VIB_{0.1}$ | **92.7** | **2.3** (3.2) |
| $VIB_{0.5}$ | 90.0 | **2.3** (2.4) |
| CEB | **92.9** | **2.3 (2.3)** |

## 6 MNI Optimality of CEB

In this work we do not attempt to give a formal proof that CEB representations learn the optimal information about the observed data (and certainly the variational form of the objective will prevent that from happening in general cases). However, CEB's targeting of the MNI is motivated by the following simple observations: If $I(X; Z) < I(X; Y)$, then we have thrown out relevant information in $X$ for predicting $Y$. If $I(X; Z) > I(X; Y)$, then we are including information in $X$ that is not useful for predicting $Y$. Thus $I(X; Z) = I(X; Y)$ is the "correct" amount of information, which is one of the equalities required in order to satisfy the MNI criterion. Only models that successfully learn that amount of information can possibly be MNI-optimal.

The second condition of MNI (Equation (3)) is only fully satisfied when optimizing the bidirectional CEB objective, described in Appendix E.2, as $\langle \log e(z_X|x) \rangle - \langle \log b(z_X|y) \rangle$ and $\langle \log b(z_Y|y) \rangle - \langle \log e(z_Y|x) \rangle$ are both 0 only when $b(z|y) = p(z|y)$ and $e(z|x) = p(z|x)$ and the corresponding decoder terms are both maximal. We leave such models for future work.

## 7 Classification Experiments

Our primary experiments are focused on comparing the performance of otherwise identical models when we change only the objective function. Consequently, we aren't interested in demonstrating state-of-the-art results for a particular classification task. Instead, we are interested in relative differences in performance that can be directly attributed to the difference in objective.

With that in mind, we present results for classification of Fashion MNIST (Xiao et al., 2017) for five different models. The five models are: a deterministic model (*Determ*); three VIB models, with $\beta \in \{\frac{1}{2}, 10^{-1}, 10^{-2}\}$ ($VIB_{0.5}$, $VIB_{0.1}$, $VIB_{0.01}$); and a CEB model. These same models are used in the calibration, out-of-distribution, and adversarial experiments (Sections 8 to 10). Critically, all five models share the same inference architecture mapping $X$ to $Y$. See Appendices C and D for details on training and the architectures.

Since Fashion MNIST doesn't have a prespecified validation set, it offers an opportunity to test training algorithms that only look at training results, rather than relying on cross validation. To that end, the five models presented here are the first models with these hyperparameters that we trained on Fashion MNIST.[6] The learning rate for the CEB model was lowered according to the training algorithm described in Appendix C. The other four models followed the same algorithm, but instead of tracking $Re_{X/Y}$, they simply tracked their training loss. All five models were required to retain the initial learning rate of 0.001 for 40 epochs before they could begin lowering the learning rate. At no point during training did any of the models exhibit non-monotonic test accuracy, so we do not believe that this approach harmed any performance – all five models converged essentially smoothly to their final, reported performance. In spite of the dynamic learning rate schedule, all five models took approximately the same number of epochs to reach the minimum learning rate.

---

[6] Development focused on MNIST (LeCun et al., 1998) and 2 dimensional latent vectors for ease of visualization. See Figure 8 for an example 2D latent space.

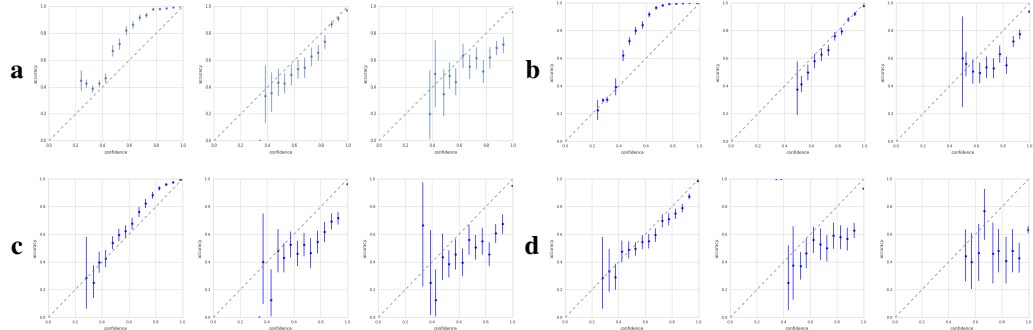

Figure 3: Calibration plots with 90% confidence intervals for four of the models after 2,000 steps, 20,000 steps, and 40,000 steps (left, center, and right of each trio, respectively): **a** is CEB, **b** is VIB$_{0.5}$, **c** is VIB$_{0.1}$, **d** is Determ. *Perfect calibration* corresponds to the dashed diagonal lines. *Underconfidence* occurs when the points are above the diagonal. *Overconfidence* occurs when the points are below the diagonal.

In the case of a simple classification problem with a uniform distribution over classes in the training set, we can directly compute $I(X; Y)$ as $\log C$, where $C$ is the number of classes.[7] See Table 1 for a comparison of the rates between the four variational models, as well as their accuracies. All but VIB$_{0.5}$ achieve the same accuracy. All four stochastic models get close to the ideal rate of 2.3 nats, but they get there by different paths. For the VIB models, the lower $\beta$ is, the higher the rate goes early in training, before converging down to (close to) 2.3 nats. CEB never goes above 2.3 nats.

## 8 CALIBRATION

In Figure 3, we show calibration plots at various points during training for the four models. Calibration curves help analyze whether models are underconfident or overconfident. Each point in the plots corresponds to a 5% confidence range. Accuracy is averaged for each bin. A *well-calibrated* model is correct half of the time it gives a confidence of 50% for its prediction.

All of the networks move from under- to overconfidence during training. However, CEB and VIB$_{0.5}$ are only barely overconfident, while $\beta = 0.1$ is sufficent to make it nearly as overconfident as the deterministic model. This overconfidence is one of the issues that is correlated with exceeding the MNI during training (Table 1). See Appendix A for a geometric explanation for how this can occur.

## 9 OUT-OF-DISTRIBUTION DETECTION

We test the ability of the five models to detect three different out-of-distribution (OoD) detection datasets. $U(0, 1)$ is uniform noise in the image domain. MNIST uses the MNIST test set. Vertical Flip is the most challenging, using vertically flipped Fashion MNIST test images, as originally proposed in Alemi et al. (2018).

We use three different metrics for thresholding. The first two, $H$ and $R$, were proposed in Alemi et al. (2018). $H$ is the classifier entropy. $R$ is the rate, defined in Section 5. The third metric is specific to CEB: $Re_{X/\hat{Y}}$. This is the predicted residual information – since we don't have access to the true value of $Y$ at test time, we use $\hat{y} \sim c(y|z_X)$ to calculate $H(Z_X|\hat{Y})$. This is no longer a valid bound on $Re_{X/Y}$, as $\hat{y}$ may not be from the true distribution $p(x, y, z_X)$. However, the better the classifier, the closer the estimate should be.

These three threshold scores are used with the standard suite of proper scoring rules: *False Positive Rate at 95% True Positive Rate* (FPR 95% TPR), *Area Under the ROC Curve* (AUROC), and *Area Under the Precision-Recall Curve* (AUPR). See Lee et al. (2018) for definitions.

---

[7]We are relying on the mild assumption that $X$ (the high-dimensional data) has higher entropy than $Y$ (the labels). $I(X; Y) \leq \min(H(X), H(Y))$.

Table 2: Results for out-of-distribution detection (*OoD*). *Thrsh.* is the threshold score used: $H$ is the entropy of the classifier; $R$ and $Re_{X/\hat{Y}}$ are defined in Section 9. Arrows denote whether higher or lower scores are better. **Bold** indicates the best score in that column for a particular OoD dataset.

| OoD | Method | Thrsh. | FPR @ 95% TPR ↓ | AUROC ↑ | AUPR In ↑ |
|---|---|---|---|---|---|
| U(0,1) | Determ | $H$ | 35.8 | 93.5 | 97.1 |
| | VIB$_{0.01}$ | $H$ | 41.1 | 92.5 | 96.0 |
| | | $R$ | **0.0** | **100.0** | **100.0** |
| | VIB$_{0.1}$ | $H$ | 43.5 | 94.5 | 96.2 |
| | | $R$ | **0.0** | **100.0** | **100.0** |
| | VIB$_{0.5}$ | $H$ | 73.2 | 87.0 | 90.5 |
| | | $R$ | 80.6 | 57.1 | 51.4 |
| | CEB | $H$ | 63.4 | 92.8 | 95.1 |
| | | $R$ | **0.0** | **100.0** | **100.0** |
| | | $Re_{X/\hat{Y}}$ | **0.0** | **100.0** | **100.0** |
| MNIST | Determ | $H$ | 59.0 | 88.4 | 90.0 |
| | VIB$_{0.01}$ | $H$ | 42.3 | 91.6 | 95.9 |
| | | $R$ | **0.0** | **100.0** | **100.0** |
| | VIB$_{0.1}$ | $H$ | 60.3 | 84.7 | 89.7 |
| | | $R$ | **0.5** | 86.8 | **99.8** |
| | VIB$_{0.5}$ | $H$ | 70.2 | 79.6 | 86.8 |
| | | $R$ | 12.3 | 66.7 | 91.1 |
| | CEB | $H$ | 70.6 | 77.8 | 73.0 |
| | | $R$ | **0.1** | 94.4 | **99.9** |
| | | $Re_{X/\hat{Y}}$ | **0.2** | 92.0 | **99.9** |
| Vertical Flip | Determ | $H$ | 66.8 | 88.6 | 90.2 |
| | VIB$_{0.01}$ | $H$ | 57.6 | 82.6 | 80.3 |
| | | $R$ | **0.0** | **100.0** | **100.0** |
| | VIB$_{0.1}$ | $H$ | 65.3 | 84.5 | 85.2 |
| | | $R$ | **0.0** | **99.2** | **100.0** |
| | VIB$_{0.5}$ | $H$ | 79.7 | 79.8 | 81.4 |
| | | $R$ | 17.3 | 52.7 | 91.3 |
| | CEB | $H$ | 68.0 | 84.9 | 85.5 |
| | | $R$ | **0.0** | 90.7 | **100.0** |
| | | $Re_{X/\hat{Y}}$ | **0.0** | 92.6 | **100.0** |

The core result is that VIB$_{0.5}$ performs much less well at the OoD tasks than the other two VIB models and CEB. We believe that this is another result of VIB$_{0.5}$ learning the right amount of information, but not learning all of the *right* information, thereby demonstrating that it is not a valid MNI objective, as explored in Appendix A. On the other hand, the other two VIB objectives seem to perform extremely well, which is the benefit they get from capturing a bit more information about the training set. We will see below that there is a price for that information, however.

## 10  ADVERSARIAL EXAMPLE ROBUSTNESS AND DETECTION

Adversarial examples were first noted in Szegedy et al. (2013). The first practical attack, *Fast Gradient Method* (FGM) was introduced shortly after (Goodfellow et al., 2015). Since then, many new attacks have been proposed. Most relevant to us is the Carlini-Wagner (CW) attack (Carlini & Wagner, 2017b), which was the first practical attack to directly use a blackbox optimizer to find minimal perturbations.[8] Many defenses have also been proposed, but almost all of them are broken (Carlini & Wagner, 2017a; Athalye et al., 2018). This work may be seen as a natural continuation of the adversarial analysis of Alemi et al. (2017), which showed that VIB naturally had robustness to whitebox adversaries, including CW. In that work, the authors did not train any VIB models with a learned $m(z_X)$, which results in much weaker models, as shown in Alemi et al. (2018). We believe this is the first work that trains a VIB model with a learned marginal and using it in an adversarial setting.

---

[8]Szegedy et al. (2013) initially used L-BFGS to find the adversaries. Carlini & Wagner (2017b) showed that it was possible to use Adam (Kingma & Ba, 2015), which is much faster.

Table 3: Results for adversarial example detection (*Attack*). All attacks are targeting the "trousers" class in Fashion MNIST. *CW* is Carlini & Wagner (2017b). *CW, (C = 1)* is CW with an additional confidence penalty set to 1. *CW, (C = 1) Det.* is a custom CW attack targeting CEB's detection mechanism, $Re_{X/\hat{Y}}$. $L_0, L_1, L_2, L_\infty$ report the corresponding norm (mean ±1 std.) of successful adversarial perturbations. Higher norms on CW indicate that the attack had a harder time finding adversarial perturbations, since it starts by looking for the smallest possible perturbation. The remaining columns are as in Table 2. Arrows denote whether higher or lower scores are better. **Bold** indicates the best score in that column for a particular adversarial attack.

| Attack | Model | Attack Success ↓ | $L_0$ ↑ | $L_1$ ↑ | $L_2$ ↑ | $L_\infty$ ↑ | Thrsh. | FPR @ 95% TPR ↓ | AUROC ↑ | AUPR In ↑ |
|---|---|---|---|---|---|---|---|---|---|---|
| CW | Determ | 100.0% | 377.1 ±100.3 | 16.2 ±10.2 | 1.4 ±1.7 | 0.2 ±0.1 | $H$ | 15.4 | 90.7 | 86.0 |
| | VIB$_{0.01}$ | 55.2% | 389.6 ±100.9 | 17.1 ±10.3 | 1.5 ±1.8 | 0.2 ±0.1 | $H$ | 11.2 | 59.9 | 90.0 |
| | | | | | | | | $R$ | **0.0** | **100.0** | **100.0** |
| | VIB$_{0.1}$ | 68.8% | 392.1 ±101.6 | 29.2 ±18.1 | 5.1 ±7.5 | 0.4 ±0.2 | $H$ | 16.5 | 77.4 | 80.0 |
| | | | | | | | | $R$ | **0.0** | **100.0** | **100.0** |
| | VIB$_{0.5}$ | **35.8%** | **432.0** ±99.6 | **40.1** ±32.1 | **9.4** ±14.4 | **0.5** ±0.3 | $H$ | 64.2 | 62.5 | 55.3 |
| | | | | | | | | $R$ | **0.0** | 98.7 | **100.0** |
| | CEB | **35.8%** | 416.4 ±97.7 | 33.6 ±30.3 | 7.4 ±15.0 | 0.3 ±0.2 | $H$ | 62.2 | 65.2 | 57.1 |
| | | | | | | | | $R$ | **0.0** | **99.7** | **100.0** |
| | | | | | | | | $Re_{X/\hat{Y}}$ | **0.0** | **99.5** | **100.0** |
| CW (C = 1) | Determ | 100.0% | 378.7 ±100.3 | 16.6 ±10.4 | 1.4 ±1.9 | 0.2 ±0.1 | $H$ | 17.9 | 90.9 | 85.7 |
| | VIB$_{0.01}$ | 96.7% | 381.3 ±101.5 | 17.4 ±10.5 | 1.6 ±1.9 | 0.2 ±0.1 | $H$ | 19.6 | 72.1 | 89.6 |
| | | | | | | | | $R$ | **0.0** | **100.0** | **100.0** |
| | VIB$_{0.1}$ | 97.3% | 382.8 ±100.4 | 28.2 ±17.2 | 4.8 ±7.4 | **0.4** ±0.2 | $H$ | 28.7 | 86.0 | 79.1 |
| | | | | | | | | $R$ | **0.0** | **100.0** | **100.0** |
| | VIB$_{0.5}$ | 50.4% | **422.0** ±101.3 | **36.4** ±28.6 | **7.8** ±12.3 | **0.4** ±0.2 | $H$ | 86.5 | 59.8 | 54.1 |
| | | | | | | | | $R$ | **0.1** | 96.2 | **100.0** |
| | CEB | **48.0%** | **417.6** ±95.5 | 33.3 ±29.8 | 7.3 ±15.4 | **0.4** ±0.2 | $H$ | 77.4 | 63.5 | 56.4 |
| | | | | | | | | $R$ | **0.0** | **99.3** | **100.0** |
| | | | | | | | | $Re_{X/\hat{Y}}$ | **0.0** | 98.7 | **100.0** |
| CW (C = 1) Det. | CEB | **25.1%** | 416.4 ±92.2 | **84.1** ±44.0 | **34.4** ±22.8 | **0.9** ±0.1 | $H$ | 95.1 | 56.4 | 45.0 |
| | | | | | | | | $R$ | 66.5 | 69.3 | 88.5 |
| | | | | | | | | $Re_{X/\hat{Y}}$ | 72.9 | 69.9 | 87.6 |

We consider CW in the whitebox setting to be the current gold standard attack, even though it is more expensive than FGM or the various iterative attacks like DeepFool (Moosavi-Dezfooli et al., 2016) or iterative variants of FGM (Kurakin et al., 2016). Running an optimizer directly on the model to find the perturbation that can fool that model tells us much more about the robustness of the model than approaches that focus on attack efficiency. CW searches over the space of perturbation magnitudes, which makes the attack hard to defend against, and thus a strong option for testing robustness.

Here, we explore three variants of the CW $L_2$ targeted attack. The implementation the first two CW attacks are from Papernot et al. (2018). CW and CW (C = 1) are the baseline CW attack, and CW with a confidence adjustment of 1. Note that in order for these attacks to succeed at all on CEB, we had to increase the default CW learning rate to $5 \times 10^{-1}$. Without that increase, CW found almost no adversaries in our early experiments. All other parameters are left at their defaults for CW, apart from setting the clip ranges to [0, 1]. The final attack, CW (C = 1) Det. is a modified version of CW (C = 1) that additionally incorporates a detection tensor into the loss that CW minimizes. For CEB, we had it target minimizing $Re_{X/\hat{Y}}$ in order to break the network's ability to detect the attack.

All of the attacks are targeting the *trouser* class of Fashion MNIST, as that is the most distinctive class. Targeting a less distinctive class, such as one of the shirt classes, would confuse the difficulty of classifying the different shirts and the robustness of the model to adversaries. We run each of the first three attacks on the *entire* Fashion MNIST test set (all 10,000 images). For the stochastic networks, we permit 32 encoder samples and take the mean classification result (the same number of samples is also used for gradient generation in the attacks to be fair to the attacker). CW is expensive, but we are able to run these on a single GPU in about 30 minutes. However, CW (C = 1) Det. ends up being

about 200 times more expensive – we were only able to run 1000 images and only 8 encoder samples, and it took $2\frac{1}{2}$ hours. Consequently, we only run CW ($C = 1$) Det. on the CEB model.

Our metric for robustness is the following: we count the number of adversarial examples that change a correct prediction to an incorrect prediction of the target class, and divide by the number of correct predictions the model makes on the non-adversarial inputs. We additionally measure the size of the resulting perturbations using the $L_0$, $L_1$, $L_2$, and $L_\infty$ norms. For CW, a larger perturbation generally indicates that the attack had to work harder to find an adversarial example, making this a secondary indication of robustness. Finally, we measure adversarial detection using the same thresholding techniques from Table 2.

The results of these experiments are in Table 3. We show all 20,000 images for four of the models in Figure 9. The most striking pattern in the models is how well $VIB_{0.01}$ and $VIB_{0.1}$ do at detection, while $VIB_{0.5}$ is dramatically more robust. We think that this is the most compelling indication of the importance of not overshooting $I(X; Y)$ – even minor amounts of overshooting appear to destroy the robustness of the model. On the other hand, $VIB_{0.5}$ has a hard time with detection, which indicates that, while it has learned a highly compressed representation, it has not learned the optimal set of bits. Thus, as we discuss in Appendix A, VIB trades off between learning the *necessary* information, which allows it to detect attacks perfectly, and learning the *minimum* information, which allows it to be robust to attacks.

The CEB model permits both – it maintains the necessary information for detecting powerful whitebox attacks, but also retains the minimum information, providing robustness. This is again visible in the CW ($C = 1$) Det. attack, which directly targets CEB's detection mechanism. Even though it no longer does well detecting the attack, the model becomes *more* robust to the attack, as indicated both by the much lower attack success rate and the much larger perturbation magnitudes.

## 11 Information-Free Generalization Experiments

We replicate the basic experiment from Zhang et al. (2016): we use the images from Fashion MNIST, but replace the training labels with fixed random labels. This dataset is information-free in the sense that $I(X; Y) = 0$. We use that dataset to train multiple deterministic models, CEB models, and a range of VIB models. We find that the CEB model *never* learns (even after 100 epochs of training), the deterministic model always learns (after about 40 epochs of training it begins to memorize the random labels), and the VIB models only learn with $\beta \leq 0.001$.

The fact that CEB and VIB with $\beta$ near $\frac{1}{2}$ manage to resist memorizing random labels is our final empirical demonstration that MNI is a powerful criterion for objective functions.

## 12 Conclusion

We have presented the basic form of the Conditional Entropy Bottleneck (CEB), motivated by the Minimum Necessary Information (MNI) criterion for optimal representations. We have shown through careful experimentation that simply by switching to CEB, you can expect substantial improvements in OoD detection, adversarial example detection and robustness, calibration, and generalization. Additionally, we have shown that it is possible to get all of these advantages without using any additional form of regularization, and without any new hyperparameters. We have argued empirically that objective hyperparameters can lead to hard-to-predict suboptimal behavior, such as memorizing random labels, or reducing robustness to adversarial examples. In Appendix E and in future work, we will show how to generalize CEB beyond the simple case of two observed variables.

It is our perspective that all of the issues explored here – miscalibration, failure at OoD tasks, vulnerability to adversarial examples, and dataset memorization – stem from the same underlying issue, which is retaining too much information about the training data in the learned representation. We believe that the MNI criterion and CEB show a path forward for many tasks in machine learning, permitting fast, amortized inference while ameliorating major problems.

Acknowledgments

REDACTED

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

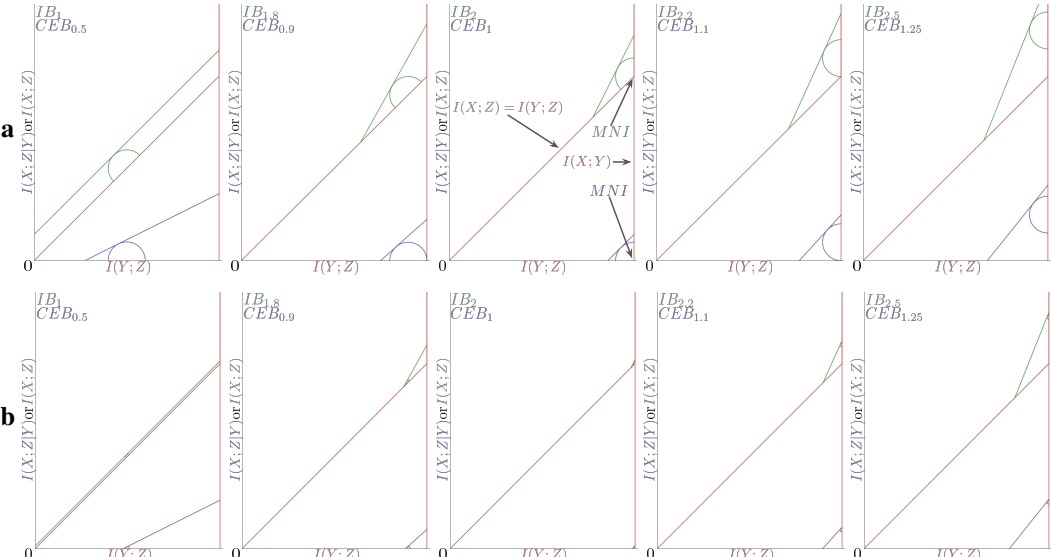

Figure 4: Geometry of the optimal surfaces for both CEB (purple) and IB (green) for models that can only come within $\epsilon$ of the optimal surface (**a**: $\epsilon = 0.1I(X;Y)$; **b**: $\epsilon = 0.01I(X;Y)$). The tangent lines have the slope of the corresponding $\beta$ – the tangent point on the $\epsilon$ ball corresponds to the point on the pareto-optimal frontier for the corresponding model. Note that $\beta$ determines the "exchange rate" between bits of $I(X;Z)$ and $I(Y;Z)$, which is how we determine the coordinate of the center of the $\epsilon$ ball. For IB to achieve the MNI point, 2 bits of $I(Y;Z)$ are needed for every bit of $I(X;Z)$. Consequently, even for an infitely powerful model (corresponding to $\epsilon = 0$), the only value of $\beta$ that hits the MNI point is $\beta = 2$. Thus, knowing the function $\epsilon(\beta)$ for a given model and dataset completely determines the model's pareto-optimal frontier.

Here we collect a number of results that are not critical to the core of the paper, but may be of interest to particular audiences.

## A  ANALYSIS OF CEB AND IB

From Equation (5) and the definition of CEB in Equation (6), the following equivalence between CEB and IB is obvious:

$$CEB \equiv I(X;Z|Y) - I(Y;Z) = I(X;Z) - 2I(Y;Z) \equiv IB_2 \tag{22}$$

where we are parameterizing IB with $\beta$ on the $I(Y;Z)$ term for convenience. This equivalence generalizes as follows:

$$IB = I(X;Z) - \beta I(Y;Z) \tag{23}$$

$$CEB = I(X;Z|Y) - \frac{\beta}{2}I(Y;Z) \tag{24}$$

In Figure 4, we show the combined information planes for CEB and IB given the above parameterization. The figures show the simple geometry that determines a point on the pareto-optimal frontier for both objectives. Every such point is fully determined by the function $\epsilon(\beta)$ for a given model and dataset, where $\epsilon$ is the closest the model can approach the true optimal surface. $\epsilon(\beta) = 0$ corresponds to the "infinite" model family that exactly traces out the boundaries of the feasible region. The full feasible regions can be seen in Figure 2.

From this geometry we can immediately conclude that if an IB model and a CEB model have the same value of $\epsilon > 0$ at equivalent $\beta$, the CEB model will always yield a value of $I(Y;Z)$ closer to $I(X;Y)$. This is because the slope of the tangent lines for CEB are always lower, putting the tangent points higher on the $\epsilon$ ball. This gives part of a theoretical justification for the empirical observations above that $VIB_{0.5}$ (equivalent to $IB_2$ in the parameterization we are describing here) fails to capture

as much of the necessary information as the CEB model. Even at the pareto-optimal frontier, $VIB_{0.5}$ cannot get $I(Y; Z)$ as close to $I(X; Y)$ as CEB can. Of course, we do not want to claim that this effect accounts for the fairly substantial difference in performance – that is likely to be due to a combination of other factors, including the fact that it is often easier to train continuous conditional distributions (like $b(z|y)$) than it is to train continuous marginal distributions (like $m(z)$).

We also think that this analysis of the geometry of IB and CEB supports our preference for targeting the MNI point and treating CEB as an objective without hyperparameters. First, there are only a maximum of 4 points of interest in both the IB and CEB information planes (all 4 are visibile in Figure 2): the origin, where there is no information in the representation; the MNI point; the point at $(I(Y; Z) = I(X; Y), I(X; Z) = H(X))$ (which is an MDL-compatible representation (Grünwald, 2007)); and the point at $(I(Y; Z) = 0, I(X; Z) = H(X|Y))$ (which would be the optimal decoder for an MNI representation). These are the only points naturally identified by the dataset – selecting a point on one of the edges between those four points seems to need additional justification. Second, if you do agree with the MNI criterion, for a given model it is impossible to get any closer to the MNI point than by setting CEB's $\beta = 1$, due to the convexity of the pareto-optimal frontier. Much more useful is making changes to the model, architecture, dataset, etc in order to make $\epsilon$ smaller. One possibility in that direction that IB and CEB models offer is inspecting training examples with high rate or residual information to check for label noise, leading to a natural human-in-the-loop model improvement algorithm. Another is using CEB's residual information as a measure of the quality of the trained model, as mentioned in Appendix C.

A final point of interest is what happens when $I(X; Y) = H(X)$. In this case, the feasible region for CEB collapses to the line segment $I(X; Z|Y) = 0$ with $0 \leq I(Y; Z) \leq I(X; Y)$. Similarly, the corresponding IB feasible region is the diagonal line $I(X; Z) = I(Y; Z)$. This case happens if we choose as our task to predict images given labels, for example. We should expect such label-conditional generative models to be particularly easy to train, since the search space is so simple. Additionally, it is never possible to learn a representation that exceeds the MNI, $I(X; Z) \leq H(X) = I(X; Y)$.

## B  MUTUAL INFORMATION OPTIMIZATION

As an objective function, CEB is independent of the methods used to optimize it. Here we focus on variational objectives because they are simple, tractable, and well-understood, but any approach to optimize mutual information terms can work, so long as they respect the side of the bounds required by the objective. For example, both Oord et al. (2018); Hjelm et al. (2018) could be used to maximize the $I(Y; Z)$ term.

There are many approaches in the literature that attempt to optimize mutual information terms in some form, including Krause et al. (2010); Chen et al. (2016); Hu et al. (2017); Hjelm et al. (2018); Oord et al. (2018). It is worth noting that none of those approaches by themselves are compatible with the MNI criterion. Some of them explicitly maximize $I(X; Z_X)$, while others maximize $I(Y; Z_X)$, but leave $I(X; Z_X)$ unconstrained. We expect all of these approaches to capture more than the MNI in general.

## C  TRAINING

Because of the properties of $Re_{X/Y}$, we can consider training algorithms that don't rely on observing validation set performance in order to decide when to lower the learning rate. The closer we can get $Re_{X/Y}$ to 0 on the training set, the better we expect to generalize to data drawn from the same distribution. One simple approach to training is to set a high initial learning rate (possibly with reverse annealing of the learning rate (Goyal et al., 2017)), and then lower the learning rate after any epoch of training that doesn't result in a new lowest mean residual information on the *training* data. This is equivalent to the logic of *dev-decay* training algorithm of Wilson et al. (2017), but does not require the use of a validation set. Additionally, since the training set is typically much larger than a validation set would be, the average loss over the epoch is much more stable, so the learning rate is less likely to be lowered spuriously. The intuition for this algorithm is that $Re_{X/Y}$ directly measures how far from optimal our learned representation is for a given $c(y|z_X)$. At the end of training $Re_{X/Y}$ indicates that we could improve performance by increasing the capacity of our architecture or

---

**Algorithm 1:** Training algorithm that lowers the learning rate when the mean $\overline{Re}_{X/Y}$ of the previous epoch is not less than the lowest $\overline{Re}^*_{X/Y}$ seen so far. The same idea can be applied to training VIB and deterministic models by tracking that the training loss is always going down. For the experiments in Section 7, we set the values specified in the **Input** section.

---

**Input :** learning_rate=$10^{-3}$, min_learning_rate=$10^{-6}$, lowering_scale=$1 - \frac{1}{e}$,
      first_epoch_when_lowering_learning_rate_is_permitted=40

---

1   epoch = 0
2   $\overline{Re}^*_{X/Y} = \infty$
3   progress = true
4   **while** *learning_rate > min_learning_rate* **do**
5     **while** *progress* **do**
       // Train and get the mean residual information.
6       $\overline{Re}_{X/Y}$ = train_model_for_1_epoch()
7       epoch = epoch +1
8       **if** $\overline{Re}_{X/Y} > \overline{Re}^*_{X/Y}$ **then**
9          progress = false
10      **else**
11          $\overline{Re}^*_{X/Y} = \overline{Re}_{X/Y}$
12     **if** *epoch ≥ first_epoch_when_lowering_learning_rate_is_permitted* **then**
13       learning_rate = learning_rate * lowering_scale
14     **else**
15       $\overline{Re}^*_{X/Y} = \infty$

---

considering ways in which our model may be misspecified. See Algorithm 1 for psuedocode. We do not claim that this algorithm is optimal.

## D  MODEL DETAILS

All of the models in our experiments have the same core architecture: A $7 \times 2$ Wide Resnet (Zagoruyko & Komodakis, 2016) for the encoder, with a final layer of $D = 4$ dimensions for the latent representation, followed by a two layer MLP classifier using ELU (Clevert et al., 2015) activations with a final categorical distribution over the 10 classes.

The stochastic models parameterize the mean and variance of a $D = 4$ fully covariate multivariate Normal distribution with the output of the encoder. Samples from that distribution are passed into the classifier MLP. Apart from that difference, the stochastic models don't differ from Determ during evaluation. None of the five models uses any form of regularization (e.g., $L_1$, $L_2$, DropOut (Srivastava et al., 2014), BatchNorm (Ioffe & Szegedy, 2015)).

The VIB models have an additional learned marginal, $m(z_X)$, which is a mixture of 240 $D = 4$ fully covariate multivariate Normal distributions. The CEB model instead has the backward encoder, $b(z_X|y)$ which is a $D = 4$ fully covariate multivariate Normal distribution parameterized by a 1 layer MLP mapping the label, $Y = y$, to the mean and variance. In order to simplify comparisons, for CEB we additionally train a marginal $m(z_X)$ identical in form to that used by the VIB models. However, for CEB, $m(z_X)$ is trained using a separate optimizer so that it doesn't impact training of the CEB objective in any way. Having $m(z_X)$ for both CEB and VIB allows us to compare the rate, $R$, of each model except Determ.

### D.1  DISTRIBUTIONAL FAMILIES

Any distributional family may be used for the encoder. Reparameterizable distributions (Kingma & Welling, 2014; Figurnov et al., 2018) are convenient, but it is also possible to use the score function trick (Williams, 1992) to get a high-variance estimate of the gradient for distributions that have no explicit or implicit reparameterization. In general, a good choice for $b(z|y)$ is the same distributional

family as $e(z|x)$, or a mixture thereof. These are modeling choices that need to be made by the practitioner, as they depend on the dataset. In this work, we chose normal distributions because they are easy to work with and will be the common choice for many problems, particularly when parameterized with neural networks, but that choice is incidental rather than fundamental.

## D.2 Regularization

Note that we did not use additional regularization on the deterministic model, but all models have a 4 dimensional bottleneck, which is likely to have acted as a strong regularizer for the deterministic model. Additionally, standard forms of regularization, including stochastic regularization, did not prevent the CW attack from being successful 100% of the time in the original work (Carlini & Wagner, 2017b). Nor did regularization cause the deterministic networks in Zhang et al. (2016) to avoid memorizing the training set. Thus, we don't think that our deterministic baseline is disadvantaged on the tasks we considered in Sections 7 and 11.

## D.3 Finitness of the Mutual Information

It is worth noting that the conditions for infinite mutual information given in Amjad & Geiger (2018) do not apply to either CEB or VIB, as they both use stochastic encoders $e(z_X|x)$. In our experiments using continuous representations, we did not encounter mutual information terms that diverged to infinity, although it is possible to make modeling and data choices that make it more likely that there will be numerical instabilities. This is not a flaw specific to CEB or VIB, however, and we found numerical instability to be almost non-existent across a wide variety of modeling and architectural choices for both variational objectives.

# E Additional CEB Objectives

Here we describe a few of the more obvious variants of the CEB objective.

## E.1 Conditional Generation

In the above presentation of CEB, we derived the objective for what may be termed "classification" tasks (although there is nothing in the derivation that restricts the form of either $X$ or $Y$). However, CEB is fully symmetric, so it is natural to consider the second task defined by our choice of dataset, conditional generation of $X$ given $Y = y$.

In this case, we can augment our graphical model with a new variable, $Z_Y$, and derive the same CEB objective for that variable:

$$\min I(Y; Z_Y|X) = \min I(Y; Z_Y) - I(X; Z_Y) \tag{25}$$
$$\Rightarrow \min -H(Z_Y|Y) + H(Z_Y|X) \tag{26}$$
$$\tag{27}$$
$$\max I(X; Z_Y) = \max H(X) - H(X|Z_Y) \tag{28}$$
$$\Rightarrow \max -H(X|Z_Y) \tag{29}$$

In the same manner as above, we can derive variational bounds on $H(Z_Y|X)$ and $H(X|Z_Y)$. In particular, we can variationally bound $p(z_Y|x)$ with $e(z_Y|x)$. Additionally, we can bound $p(x|z_Y)$ with a decoder distribution of our choice, $d(x|z_Y)$.

Because the decoder is maximizing a lower bound on the mutual information between $Z_Y$ and $X$, it can never memorize $X$. It is directly limited during training to use exactly $H(Y)$ nats of information from $Z_Y$ to decode $X$. For a mean field decoder, this means that the decoder will only output a canonical member of each class. For a powerful decoder, such as an autoregressive decoder, it will learn to select a random member of the class.

For discrete $Y$, this model can trivially be turned into an unconditional generative model by first sampling $Y$ from the training data or using any other appropriate procedure, such as sampling $Y$ uniformly at random.

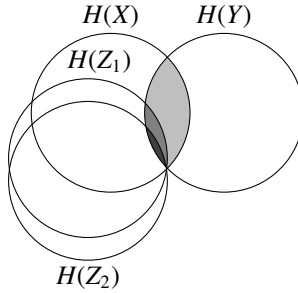

Figure 5: Information diagram for the basic hierarchical CEB model, $Z_2 \leftarrow Z_1 \leftarrow X \leftrightarrow Y$.

### E.2 Bidirectional Generation

Given the presentation of conditional generation above, it is natural to consider that both $c(y|z)$ and $d(x|z)$ are conditional generative models of $Y$ and $X$, respectively, and to learn a $Z$ that can handle both tasks. This can be done easily with the following bidirectional CEB model: $Z_X \leftarrow X \leftrightarrow Y \rightarrow Z_Y$. This corresponds to the following factorization: $p(x, y, z_X, z_Y) \equiv p(x, y)e(z_X|x)b(z_Y|y)$. The two objectives from above then become the following single objective:

$$\min \; -H(Z_X|X) + H(Z_X|Y) + H(Y|Z_X) \tag{30}$$
$$- H(Z_Y|Y) + H(Z_Y|X) + H(X|Z_Y) \tag{31}$$

A natural question is how to ensure that $Z_X$ and $Z_Y$ are consistent with each other. Fortunately, that consistency is trivial to encourage by making the natural variational approximations: $p(z_Y|x) \rightarrow e(z_Y|x)$ and $p(z_X|y) \rightarrow b(z_X|y)$. The full bidirection variational CEB objective then becomes:

$$\min \; \langle \log e(z_X|x) \rangle - \langle \log b(z_X|y) \rangle - \langle \log c(y|z_X) \rangle$$
$$+ \langle \log b(z_Y|y) \rangle - \langle \log e(z_Y|x) \rangle - \langle \log d(x|z_Y) \rangle \tag{32}$$

At convergence, we learn a unified $Z$ that is consistent with both $Z_X$ and $Z_Y$, permitting generation of either output given either input in the trained model, in the same spirit as Vedantam et al. (2018), but without any objective function hyperparameter tuning.

### E.3 Hierarchical CEB

Thus far, we have focused on learning a single latent representation (possibly composed of multiple latent variables at the same level). Here, we consider how to learn a hierarchical model with CEB.

Consider the graphical model $Z_2 \leftarrow Z_1 \leftarrow X \leftrightarrow Y$. This is the simplest hierarchical supervised representation learning model. The general form of its information diagram is given in Figure 5.

The key observation for generalizing CEB to hierarchical models is that the target mutual information doesn't change. By this, we mean that all of the $Z_i$ in the hierarchy should cover $I(X; Y)$ at convergence, which means maximizing $I(Y; Z_i)$. It is reasonable to ask why we would want to train such a model, given that the final set of representations are presumably all effectively identical in terms of information content. The answer is simple: doing so allows us to train deep models in a principled manner such that all layers of the network are consistent with each other and with the data. We need to be more careful when considering the residual information terms, though – it is not the case that we want to minimize $I(X; Z_i|Y)$, which is not consistent with the graphical model. Instead, we want to minimize $I(Z_{i-1}; Z_i|Y)$, defining $Z_0 = X$.

This gives the following simple *Hierarchical CEB* objective:

$$CEB_{\text{hier}} \equiv \min \sum_i I(Z_{i-1}; Z_i|Y) - I(Y; Z_i) \tag{33}$$

$$\Leftrightarrow \min \sum_i -H(Z_i|Z_{i-1}) + H(Z_i|Y) + H(Y|Z_i) \tag{34}$$

Because all of the $Z_i$ are targeting $Y$, this objective is as stable as regular CEB. Note that if all of the $Z_i$ have the same dimensionality, in principle they may all use the same networks for $b(z_i|Y)$ and/or $c(y|z_i)$, which may substantially reduce the number of parameters in the model. All of the individual loss terms in the objective must still appear, of course. There is no requirement, however, that the $Z_i$ have the same latent dimensionality, although doing so may give a unified hiearchical representation.

### E.4 Sequence Learning

Many of the richest problems in machine learning vary over time. In Bialek & Tishby (1999), the authors define the *Predictive Information*:

$$I(X_{past}, X_{future}) = \left\langle \log \frac{p(x_{past}, x_{future})}{p(x_{past})p(x_{future})} \right\rangle$$

This is of course just the mutual information between the past and the future. However, under an assumption of temporal invariance (any time of fixed length is expected to have the same entropy), they are able to characterize the predictive information, and show that it is a subextensive quantity: $\lim_{T \to \infty} I(T)/T \to 0$, where $I(T)$ is the predictive information over a time window of length $2T$ ($T$ steps of the past predicting $T$ steps into the future). This concise statement tells us that past observations contain vanishingly small information about the future as the time window increases.

The application of CEB to extracting the predictive information is straightforward. Given the Markov chain $X_{<t} \to X_{\geq t}$, we learn a representation $Z_t$ that optimally covers $I(X_{<t}, X_{\geq t})$ in *Predictive CEB*:

$$CEB_{\text{pred}} \equiv \min I(X_{<t}; Z_t | X_{\geq t}) - I(X_{\geq t}, Z_t) \tag{35}$$

$$\Rightarrow \min -H(Z_t | X_{<t}) + H(Z_t | X_{\geq t}) + H(X_{\geq t} | Z_t) \tag{36}$$

Note that the model entailed by this objective function *does not* rely on $Z_{<t}$ when predicting $X_{\geq t}$. A single $Z_t$ captures all of the information in $X_{<t}$ and is to be used to predict as far forward as is desired. "Rolling out" $Z_t$ to make predictions is a modeling error according to the predictive information.

Also note that, given a dataset of sequences, $CEB_{\text{pred}}$ may be extended to a bidirectional model, as in Appendix E.2. In this case, two representations are learned, $Z_{<t}$ and $Z_{\geq t}$. Both representations are for timestep $t$, the first representing the observations before $t$, and the second representing the observations from $t$ onwards. As in the normal bidirectional model, using the same encoder and backwards encoder for both parts of the bidirectional CEB objective ties the two representations together.

**Modeling and architectural choices.**  As with all of the variants of CEB, whatever entropy remains in the data after capturing the entropy of the mutual information in the representation must be modeled by the decoder. In this case, a natural modeling choice would be a probalistic RNN with powerful decoders per time-step to be predicted. However, it is worth noting that such a decoder would need to sample at each future step to decode the subsequent step. An alternative, if the prediction horizon is short or the predicted data are small, is to decode the entire sequence from $Z_t$ in a single, feed-forward network (possibly as a single autoregression over all outputs in some natural sequence). Given the subextensivity of the predictive information, that may be a reasonable choice in stochastic environments, as the useful prediction window may be small.

**Multi-scale sequence learning.**  As in WaveNet (Van Den Oord et al., 2016), it is natural to consider sequence learning at multiple different temporal scales. Combining an architecture like time-dilated WaveNet with CEB is as simple as combining $CEB_{\text{pred}}$ with $CEB_{\text{hier}}$ (Appendix E.3). In this case, each of the $Z_i$ would represent a wider time dilation conditioned on the aggregate $Z_{i-1}$. The advantage of such an objective over that used in WaveNet is avoiding unnecessary memorization of earlier timesteps.

### E.5 Unsupervised CEB

Pure unsupervised learning is fundamentally an ill-posed problem. Without knowing what the task is, it is impossible to define an optimal representation directly. We think that this core issue is what lead the authors of Bengio et al. (2013) to prefer barely compressed representations. But by that line of

reasoning, it seems that unsupervised learning devolves to lossless compression – perhaps the correct representation is the one that allows you to answer the question: "What is the color of the fourth pixel in the second row?"

On the other hand, it also seems challenging to put the decision about what information should be kept into objective function hyperparameters, as in the $\beta$ VAE and penalty VAE (Alemi et al., 2018) objectives. That work showed that it is possible to constrain the amount of information in the learned representation, but it is unclear how those objective functions keep only the "correct" bits of information for the downstream tasks you might care about. This is in contrast to all of the preceeding discussion, where the task clearly defines the both the correct amount of information and which bits are likely to be important.

However, unsupervised representation learning is still an interesting problem, even if it is ill-posed. Our perspective on the importance of defining a task in order to constrain the information in the representation suggests that we can turn the problem into a data modeling problem in which the practitioner who selects the dataset also "models" the likely form of the useful bits in the dataset for the downstream task of interest.

In particular, given a dataset $X$, we propose selecting a function $f(X) \rightarrow X'$ that transforms $X$ into a new random variable $X'$. This defines a paired dataset, $P(X, X')$, on which we can use CEB as normal. Note that choosing the identity function for $f$ results in maximal mutual information between $X$ and $X'$ ($H(X)$ nats), which will result in a representation that is far from the MNI for normal downstream tasks. In other words, representations learned by true autoencoders are unlikely to be any better than simply using the raw $X$.

It may seem that we have not proposed anything useful, as the selection of $f(.)$ is unconstrained, and seems much more daunting than selecting $\beta$ in a $\beta$ VAE or $\sigma$ in a penalty VAE. However, there is a very powerful class of functions that makes this problem much simpler, and that also make it clear using CEB will *only* select bits from $X$ that are useful. That class of functions is the noise functions.

### E.5.1 Denoising CEB Autoencoder

Given a dataset $X$ without labels or other targets, and some set of tasks in mind to be solved by a learned representation, we may select a random noise variable $U$, and function $X' = f(X, U)$ that we believe will destroy the irrelevant information in $X$. We may then add representation variables $Z_X, Z_{X'}$ to the model, giving the joint distribution $p(x, x', u, z_X, z_{X'}) \equiv p(x)p(u)p(x'|f(x, u))e(z_X|x)b(z_{X'}|x')$. This joint distribution is represented in Figure 6.

*Denoising Autoencoders* were originally proposed in Vincent et al. (2008). In that work, the authors argue informally that reconstruction of corrupted inputs is a desirable property of learned representations. In this paper's notation, we could describe their proposed objective as $\min H(X|Z_{X'})$, or equivalently $\min \langle \log d(x|z_{X'} = f(x, \eta)) \rangle_{x, \eta \sim p(x)p(\theta)}$.

Here we make this idea somewhat more formal through the MNI criterion and the derivation of CEB as the optimal objective for that criterion. We also note that, practically speaking, we would like to learn a representation that is consistent with uncorrupted inputs as well. Consequently, we are going to use a bidirectional model.

$$CEB_{\text{denoise}} \equiv \min I(X; Z_X|X') - I(X'; Z_X) + I(X'; Z_{X'}|X) - I(X; Z_{X'}) \tag{37}$$
$$\Rightarrow \min -H(Z_X|X) + H(Z_X|X') + H(X'|Z_X) - H(Z_{X'}|X') + H(Z_{X'}|X) + H(X|Z_{X'}) \tag{38}$$

This requires two encoders and two decoders, which may seem expensive, but it permits a consistent learned representation that can be used cleanly for downstream tasks. Using a single encoder/decoder pair would result in either an encoder that does not work well with uncorrupted inputs, or a decoder that only generates noisy outputs.

If you are only interested in the learned representation and not in generating good reconstructions, the objective simplifies to the first three terms. In that case, the objective is properly called a *Noising CEB Autoencoder*, as the model predicts the noisy $X'$ from $X$:

$$CEB_{\text{noise}} \equiv \min I(X; Z_X|X') - I(X'; Z_X) \tag{39}$$
$$\Rightarrow \min -H(Z_X|X) + H(Z_X|X') + H(X'|Z_X) \tag{40}$$

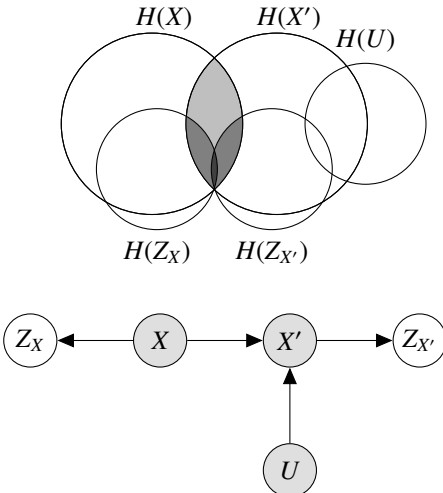

Figure 6: Information diagram and graphical model for the Denoising CEB Autoencoder.

In these models, the noise function, $X' = f(X, U)$ must encode the practitioner's assumptions about the structure of information in the data. This obviously will vary per type of data, and even per desired downstream task.

However, we don't need to work too hard to find the perfect noise function initially. A natural first choice for $f$ is:[9]

$$f(x, \eta) = \text{clip}(x + \eta, \mathcal{D}) \tag{41}$$

$$\eta \sim \lambda U(-1, 1) * \mathcal{D} \tag{42}$$

$$\mathcal{D} = \text{domain}(X) \tag{43}$$

In other words, add uniform noise scaled to the domain of $X$ and by a hyperparameter $\lambda$, and clip the result to the domain of $X$. When $\lambda = 1$, $X'$ is indistinguishable from uniform noise. As $\lambda \to 0$, this maintains more and more of the original information from $X$ in $X'$. For some value of $\lambda > 0$, most of the irrelevant information is destroyed and most of the relevant information is maintained, if we assume that higher frequency content in the domain of $X$ is less likely to contain the desired information. That information is what will be retained in the learned representation.

**Theoretical optimality of noise functions.** Above we claimed that this learning procedure will only select bits that are useful for the downstream task, given that we select the proper noise function. Here we prove that claim constructively. Imagine an oracle that knows which bits of information should be destroyed, and which retained in order to solve the future task of interest. Further imagine, for simplicity, that the task of interest is classification. What noise function must that oracle implement in order to ensure that $CEB_{denoise}$ can only learn exactly the bits needed for classification? The answer is simple: for every $X = x_i$, select $X' = x'_i$ uniformly at random from among all of the $X = x_j$ that should have the same class label as $X = x_i$. Now, the only way for CEB to maximize $I(X; Z_{X'})$ and minimize $I(X'; Z_{X'})$ is by learning a representation that is isomorphic to classification, and that encodes exactly $I(X; Y)$ nats of information, even though it was only trained "unsupervisedly" on $X, X'$ pairs. Thus, if we can choose the correct noise function that destroys only the bits we don't care about, $CEB_{denoise}$ will learn the desired representation and nothing else (caveated by model, architecture, and optimizer selection, as usual).

### E.6 SEMI-SUPERVISED CEB

Given *any* amount of paired data $X, Y$ immediately improves our ability to learn a semantic representation. Fortunately, it is easy to reincorporate paired data in combination with noising and

---

[9]White noise is probably a better choice for audio signals, and may be the right choice for most real-valued signals, including images and videos.

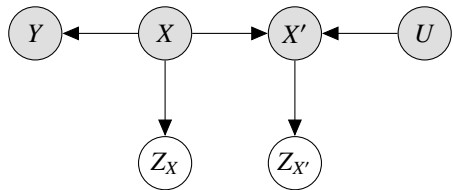

Figure 7: Graphical model for Semi-Supervised CEB.

denoising CEB, introduced above. We present the assumed graphical model in Figure 7. We give the corresponding *Semi-Supervised CEB* directly:

$$CEB_{semi} \equiv \min I(X; Z_X|X') - I(X'; Z_X) + I(X'; Z_{X'}|X) - I(X; Z_{X'}) \tag{44}$$

$$+ \mathbf{1}_{Y \in (X,Y)}[I(X'; Z_{X'}|Y) - I(Y; Z_{X'})] \tag{45}$$

$$\Rightarrow \min -H(Z_X|X) + H(Z_X|X') + H(X'|Z_X) - H(Z_{X'}|X') + H(Z_{X'}|X) + H(X|Z_{X'}) \tag{46}$$

$$+ \mathbf{1}_{Y \in (X,Y)}[-H(Z_{X'}|Y) + H(Z_{X'}|Y) + H(Y|Z_{X'})] \tag{47}$$

$\mathbf{1}_{Y \in (X,Y)}$ is the indicator function, equal to 1 when a $Y$ is part of the paired data, and equal to 0 otherwise. In other words, if we have $Y = y$ paired with a given $X = x$, we can include those terms in the objective. If we do not have that, we can simply leave them out.

Note that it is straightforward to generalize this to semisupervised learning with two or more observations that are both being learned unsupervisedly, but also have some amount of paired data. For example, images and natural language, assuming we have a reasonable noise model for unsupervisedly learning natural language.

## F  VISUALIZATIONS

Here we provide some visualizations of the Fashion MNIST tasks.

In Figure 8, we show a trained 2D CEB latent representation of Fashion MNIST. The model learned to locate closely related concepts together, including the cluster of "shirt" classes near the center, and the cluster of "shoe" classes toward the lower right. In spite of the restriction to 2 dimensions, this model achieves $\sim 92\%$ on the test set.

In Figure 9, the 10,000 test images and their 10,000 adversaries are shown for four of the models. It is easy to see at a glance that the CEB model organizes all of the adversaries into the "trousers" class, with a crisp devision between the true examples and the adversaries. In contrast, the two VIB models have adversaries mixed throughout. However, all three models are clearly preferable to the deterministic model, which has all of the adversaries mixed into the "trousers" class with no ability to distinguish between adversaries and true examples.

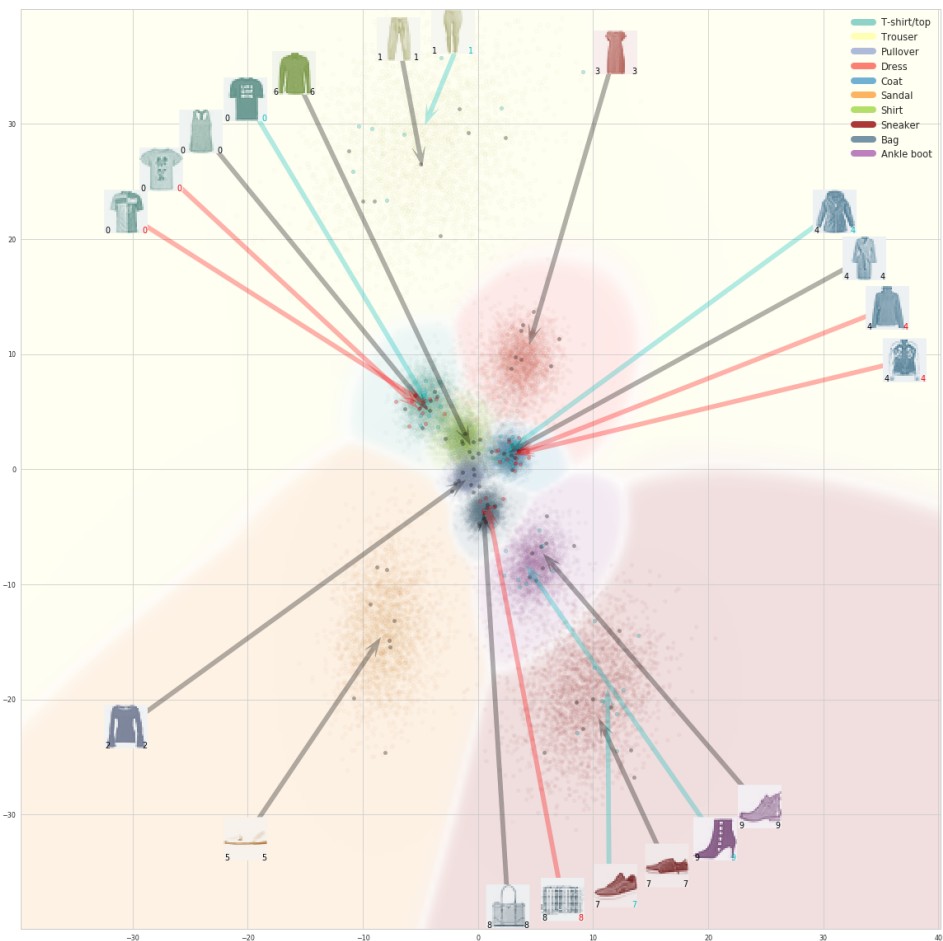

Figure 8: A trained 2D latent space for a Fashion MNIST CEB model.

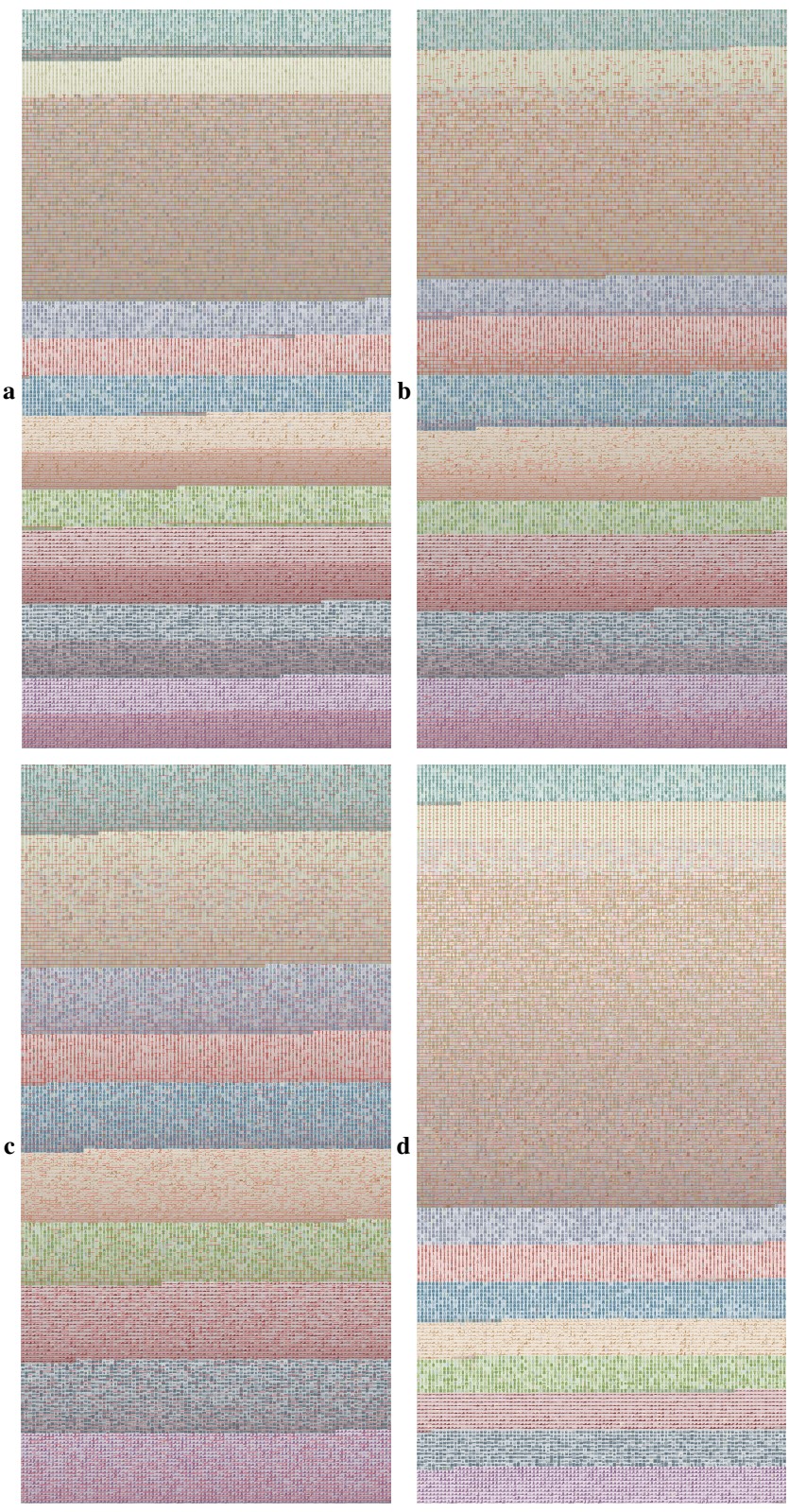

Figure 9: All adversarial images sorted by predicted class, showing the difference between robust and non-robust models. Each predicted class is sorted by the model's rate, $R$ ($H$ is used for **d**), from low to high. Images with a red bar along their top are adversarial. **a** is CEB, **b** is $VIB_{0.5}$, **c** is $VIB_{0.01}$, **d** is Determ.

