# OpenReview forum: "The Conditional Entropy Bottleneck"
_ICLR.cc/2019/Conference_

### Official Review · AnonReviewer2 · 2018-11-02
**Interesting approach with decent results, but far lacking in related works on mutual information**

**Rating:** 6
**Confidence:** 3

**Review:**

Update: see comments "On revisions" below.

This paper essentially introduces a label-dependent regularization to the VIB framework, matching the encoder distribution of one computed from labels. The authors show good performance in generalization, such that their approach is relatively robust in a number of tasks, such as adversarial defense.

The idea I think is generally good, but there are several problems with this work.

First, there has been recent advances in mutual information estimation, first found in [1]. This is an important departure from the usual variational approximations used in VIB. You need to compare to this baseline, as it was shown that it outperforms VIB in a similar classification task as presented in your work.

Second, far too much space is used to lay out some fairly basic formalism with respect to mutual information, conditional entropy, etc. It would be nice, for example, to have an algorithm to make the learning objective more clear. Overall, I don't feel the content justifies the length.

Third, I have some concerns about the significance of this work. They introduce essentially a label-dependent “backwards encoder” to provide samples for the KL term normally found in VIB. The justification is that we need the bottleneck term to improve generalization and the backwards encoder term is supposed to keep the representation relevant to labels. One could have used an approach like MINE, doing min information for the bottleneck and max info for the labels. In addition, much work has been done on learning representations that generalize using mutual information (maximizing instead of minimizing) [2, 3, 4, 5] along with some sort of term to improve "relevance", and this work seems to ignore / not be aware of this work.

Overall I could see some potential in this paper being published, as I think the approach is sensible, but it's not presented in the proper context of past work.

[1] Belghazi, I., Baratin, A., Rajeswar, S., Courville, A., Bengio, Y., & Hjelm, R. D. (2018). MINE: mutual information neural estimation. International Conference for Machine Learning, 2018.
[2] Gomes, R., Krause, A., and Perona, P. Discriminative clustering by regularized information maximization. In NIPS, 2010.
[3] Hu, W., Miyato, T., Tokui, S., Matsumoto, E., and Sugiyama, M. Learning discrete representations via information maximizing self-augmented training. In ICML, 2017.
[4] Hjelm, R. D., Fedorov, A., Lavoie-Marchildon, S., Grewal, K., Trischler, A., & Bengio, Y. (2018). Learning deep representations by mutual information estimation and maximization. arXiv preprint arXiv:1808.06670.
[5] Oord, Aaron van den, Yazhe Li, and Oriol Vinyals. "Representation learning with contrastive predictive coding." arXiv preprint arXiv:1807.03748 (2018).

---

> ### Comment · AnonReviewer3 · 2018-11-04
> **MINE cannot be used to minimize mutual information**
>
> MINE is a lower bound to the mutual information (and that fails too in the minibatch setting), and thus cannot be used to minimize mutual information. The approach taken by this paper yields a proper upper bound even in the minibatch setting.

---

> > ### Comment · AnonReviewer2 · 2018-11-04
> > **MINE and mutual information minimization.**
> >
> > MINE in the min max framework (information bottleneck) works exactly like GANs in that a separate discriminator (statistics network in the paper) maximizes the lower bound to the expected log ratio of the joint over the product of marginals. The encoder is optimized to minimize this estimate, which is the same as the GAN generator. I can see the advantage to VIB-type (min min) methods as adversarial objectives have many optimization difficulties, but MINE has the advantage of not needing explicit densities.
> >
> > I'm not sure what you mean by that MINE fails: MINE is demonstrated to outperform VIB by a good margin in a experiment very similar to one presented in this submission.  (See section 5.3 of MINE).
> >
> > Minibatch MINE has biased gradients, but the authors introduce a learned baseline to address this.  Please see section 3.2 in the MINE paper on bias correction.
> >
> > As many recent works have found success in MINE-like techniques and MINE has been shown to outperform VIB in a very similar setting, the submission authors would need a much more compelling argument not to include some comparison to MINE.

---

> > > ### Author Response · Authors · 2018-11-17
> > > **Concerns addressed above**
> > >
> > > We hope that our discussion of MINE in our main response above sufficiently addresses your concerns here. Please let us know if that is not the case.

---

> > ### Author Response · Authors · 2018-11-17
> > **Different optimization choices**
> >
> > This comment seems correct, but we agree with the reviewer that MINE and CPC could be used to learn a CEB model. All that is required is that those lower bounds on mutual information be used to optimize the I(Y;Z) term in the objective. CEB is an objective function, but any appropriate technique can be chosen to optimize any of its terms, such as variational approximations for the upper bound (I(X;Z|Y)) and CPC for the lower bound (I(Y;Z)). We don’t explore those here for the reasons we mention in our main response above.

---

> ### Author Response · Authors · 2018-11-17
> **Response, Part 1**
>
> Thank you for your review, and for your thorough search of the literature on uses of the mutual information in objective functions. We were aware of the work you mentioned. The high order bit is that you are correct -- any estimator of the mutual information could be used in conjunction with the CEB objective, assuming the inequalities of the estimator correspond correctly to the direction the mutual information estimate needs to be optimized.  We chose to explore variational bounds in this paper because they are simple, tractable, and well-understood, but we mention in Section 3 that other approaches are possible. We will be more specific about that in our revisions.
>
> There is quite a bit of care that needs to be taken to understand these mutual information estimators, but both MINE and CPC may be used to optimize the lower bound on I(Z;Y). For reasons we describe in more detail below, we don’t think that doing so would substantially change the results in these particular experiments, but when applying CEB to tasks other than classification, both options are worth considering, in addition to the variational approach we present. In other words, the CEB objective involves minimization and maximization of mutual information terms, and any correctly bounded method to estimate mutual information could be used to optimize the components of the general CEB objective. Our focus on the variational approach in the paper is merely convenient, rather than essential.
>
> Below we discuss MINE in more detail, but first we would like to clarify that all five of the papers you mention are proposing objectives that are not consistent with the MNI criterion if used by themselves, rather than as part of the optimization approach for an implementation of CEB. They are all focusing on maximizing estimations or lower bounds of either I(X;Z) or I(Y;Z). As we point out in Section 3, maximizing I(Y;Z) is necessary but not sufficient for achieving the Minimum Necessary Information, and maximizing I(X;Z) is fundamentally inconsistent with MNI. Additionally, it is worth pointing out that standard maximum likelihood estimation training also maximizes I(Y;Z) in deep networks, where Z can be taken as any intermediate layer of the network -- minimizing the cross entropy is the same as minimizing the H(Y|Z) term in the paper, which maximizes a lower bound on I(Y;Z) - H(Y), and H(Y) is constant with respect to the parameters. Thus, we expect that all five of these approaches (when used by themselves) still suffer from the excess of information in the representation that we hypothesize MLE to suffer from (whether the representation is explicit, as in VIB, Gomes et al., 2010, and Hjelm et al., 2018, or implicit, as in the other 3 papers you mention). In contrast, CEB is maximizing I(Y;Z) while also minimizing I(X;Z|Y), which forces optimization to get the trained model as close as it can to the MNI goal state of I(X;Z) = I(Y;Z) = I(X;Y).
>
> To summarize the point we are making here, we do not consider unconstrained maximization of the mutual information between an observed variable and a representation variable to be a desirable property by itself, and we show that doing so is inconsistent with the MNI criterion. Such techniques can only be made compatible with MNI when used to optimize a complete MNI-compatible objective function, such as CEB.

---

> > ### Author Response · Authors · 2018-11-17
> > **Response, Part 2**
> >
> > MINE
> >
> > The MINE estimator introduced an interesting set of ideas. However, it has a number of problems. First, due to the expectation inside of the log in the proposed MINE objective function, the direction of the bound is lost -- it becomes a stochastic _estimate_ of the mutual information, possibly with very high variance. During training, the authors perform an adjustment to the objective in the name of reducing variance of the gradients, but this adjustment additionally corrects the objective to again be a lower bound. In other words, the proposed MINE objective only works as a lower bound to maximize at all thanks to the gradient correction term. Second, the objective relies on taking O(K^2) passes over each minibatch of K examples, which means that the batch size is severely restricted in practice. However, the tightness of the bound relies on K being large. This is problematic. Finally, MINE appears to be much more challenging to implement and to train than variational approaches, due to the minimax nature of the estimation and the strong sensitivity to the batch size.
> >
> > As for comparisons between MINE and CEB, there are two important points to make.
> >
> > First, doing so would require using a very different architecture than the one we were able to use for all of the other models, where all three approaches we compare can be formulated in terms of an encoder and a classifier at inference time. MINE does not factor neatly into either of those pieces. This would break the apples-to-apples nature of our current set of experiments. In our mind, doing so would substantially reduce the clarity of the experimental results, while not clearly providing benefit to the core story. We hope you agree with that perspective, and that revising the paper to make it more explicit that non-variational estimators of the mutual information can be used to train CEB models will satisfy your concerns about related work.
> >
> > The second point about experimental comparisons with MINE is that the comparison with VIB presented in the MINE paper is flawed. The authors chose to use MINE to optimize I(X;Z) in the VIB objective, but since MINE (as implemented) is a lower bound and VIB is minimizing I(X;Z) rather than maximizing it, that choice is incorrect. The end result can only have been a model that tried to do as well as possible at the classification problem while also maintaining as _much_ information as possible about X, which is the opposite treatment of I(X;Z) than what the Information Bottleneck principle proposes. Given this experimental error, it does not seem appropriate to cite MINE for its empirical comparison to VIB.
> >
> > We do not mean for our comments here to be an attack on MINE, which we consider an important contribution, particularly for showing that there are more ways to estimate mutual information than had been previously considered. We will incorporate citations of both MINE and CPC in our revisions.
> >
> >
> > Length and Content
> >
> > We agree that the paper can be improved by including explicit pseudocode for the training algorithm, and the writing can definitely be tightened. Our revisions will include such changes.
> >
> > It is true that we wrote the derivation of CEB in an intentionally pedagogical manner -- many authors would skip many of the steps in the derivation. As an expert, you may have found that exposition tedious. We felt that showing this level of detail in the derivation and explaining why each choice needed to be made would help reduce the likelihood of other objectives being proposed that, for example, resulted in an incorrect bound, variational or otherwise.

---

> > ### Comment · AnonReviewer2 · 2018-11-30
> > **On revisions**
> >
> > The revision motivations are much clearer and to-the-point, and the inclusions of Figures 2 and 4 are very helpful for understanding where this method lies w.r.t. VIB.
> >
> > My primary concerns in my original review were the framing of this work as being "finding good representations", but it seems like this is really about finding good representations relevant to some other known variable (such as the labels). Since you appear to be making some broader claims w.r.t. representations, this opens you up to comparisons to methods that are arguably more general in that they operate primarily as unsupervised methods, including the self-supervision and data-augmentation-driven methods I mentioned. You are correct that they do not address MNI as directly as in your work, as each of these could encode information irrelevant for predicting some other known variable. However, as "representation learning" tools, they are far more powerful, as demonstrated in their ability to work on high-dimensional datasets.
> >
> > As a study of how to learn MI models between two known variables, X and Y, this work has a lot of value, but I would make the setting a bit more clearer in the beginning.
> >
> > I do wish that there were more datasets here, as a study with CelebA attributes or CUB captions / attributes would be very convincing.
> >
> > Your concerns about MINE are duly noted, but MINE comes with the strong advantage of not needing to specify the posterior (for instance, a noise-injected nn will work). The additional network needed is just another encoder similar to the one used in your e(z_x | x). In the IB setting, this works precisely as with GANs, except in this case the encoder tries to (adversarially) make the joint distribution resemble the product of marginals. Besides, MINE is demonstrated to work better than your baseline, so shouldn't that be of note?
> >
> > Anyways, as the revision is a bit better, I'll increase my score to a 6, but I need to read more thoroughly the other reviewers' concerns before I move any further.
> >
> > One thought on making this fully unsupervised, and possibly a stronger tool for learning representations: what about sampling X and Y from a random mask (i.e., a crop) and the corresponding negative mask on the image? Enforcing MNI would result in a latent representation which contains the information that is shared between the positive and negative masked areas, which is closer to these self-supervision methods I mentioned.

---

> > > ### Author Response · Authors · 2018-12-04
> > > **Response**
> > >
> > > Thank you for reading our revisions, and for adjusting your score. You are correct that we focus on the body of the paper on supervised representation learning. This is representation learning as presented in Tishby et al. (2000).
> > >
> > > However, we explicitly state in the main body of the paper that we are placing no restrictions on the nature of X and Y. It is entirely possible for X and Y to be the same random variable. In this case, you may choose to have e(z_X|x) be the same distribution as b(z_X|x), resulting in the objective simplifying to min -<log d(x|z_X)>, where d(x|z_X) is a decoder distribution. This is just a stochastic autoencoder, of course. Another way to put it is that the general CEB objective simplifies to max I(X;Z_X), which could be optimized using a stochastic autoencoder, MINE, or CPC. You may also choose to use two different encoder distributions (i.e., two different architectures and/or sets of parameters parameterizing the forward and backward encoders), in which case the objective is the same as presented in the main body.
> > >
> > > The immediate consequence of either of these choices is that the MNI point coincides with H(X). This is exactly the amount of information that MINE and the other unsupervised representation learning papers are targeting when they maximize I(X;Z_X). The only things that will result in I(X;Z_X) < H(X) are modeling and architecture choices that prohibit learning so much information. A sufficiently powerful model would learn I(X;Z_X) = H(X) with all such objectives, in other words.
> > >
> > > Having said all of that, we also describe in the appendix a CEB objective for unsupervised learning that gets at what we think are some of the core issues with unsupervised representation learning. We have done fairly extensive experimentation with that approach and have found it to be a substantial improvement over, for example, beta VAE as explored in Alemi et al. (2018) when measured by training a separate classifier on the unsupervised representation. We will present that approach in more detail in later work, but mention it here in order to reassure you that we are describing a general approach to representation learning. Indeed, your proposal at the end of your comment has a lot in common with the approach we describe in the appendix. Your masking suggestion corresponds to a particular choice of a noise function which serves to limit the amount of information the CEB representation will learn. We consider the particular choice of noise function to be a modeling problem, since only the practitioner will know what downstream tasks are important, and what noise is likely to destroy the information that is irrelevant for those tasks on a particular dataset (although naive choices are already quite effective in our experiments, at least when the downstream task is classification).
> > >
> > > If we understand correctly, your point about the MINE experiments in comparison with VIB is that the reported classification accuracy is better on the task using the MINE model compared to what is reported in the original VIB paper. We agree that that is the case, but we don’t think it is a compelling reason to compare to MINE in our experiments for the following reasons:
> > >  - The VIB models were weaker than ours due to the lack of a learned marginal, and consequently are likely to have had a very loose upper bound on I(X;Z) which in our experience with VIB results in a substantial decrease in performance.
> > >  - The difference in performance between MINE and VIB on the task in question is small.
> > >  - The MINE models used to compare with VIB indicate confusion about the purpose of the I(X;Z) term in the VIB objective, since (as we noted) MINE was used to maximize it rather than minimize it. We think your comment about IB and MINE is attempting to refute this point, but we don’t see how the GAN-like nature of MINE changes the fact that MINE is a lower bound on I(X;Z), while IB requires that I(X;Z) be minimized. It is not possible to minimize a term by optimizing a lower bound on that term.
> > >  - As we previously stated (and as we make clear in our revisions), CEB is an objective function that can be optimized with any valid bounds on the mutual information terms. Our experiments are explicitly designed to focus on understanding differences than can be attributed to the objective functions themselves (MLE, amortized IB, amortized CEB), rather than an exploration of all the different ways that amortized CEB can be optimized.
> > >
> > > In the interim, we have implemented CPC in order to compare its performance in CEB with mean field and pixelcnn decoders when training the bidirectional objective presented in the appendix. We agree that the strength of MINE and CPC is that they avoid the use of costly decoders like pixelcnn when maximizing mutual information with a high dimensional input like an image, and our future work will present extensive experimental comparisons with such approaches.

---

### Official Review · AnonReviewer1 · 2018-11-05
**IB with a specific choice of a parameter, but different from than VIB**

**Rating:** 2
**Confidence:** 4

**Review:**

This paper wants to discuss a new objective function, which the authors dub "Conditional Entropy Bottleneck" (CEB), motivated by learning better latent representations. However, as far as I can tell, the objective functions already exists in the one-parameter family of Information Bottleneck (IB) of Tishby, Pereira, and Bialek. The author seems to realize this in Appendix B, but calls it "a somewhat surprising theoretical result". However, if we express IB as max I(Z;Y) - beta I(Z;X), see (19), and then flip signs and take the max to the min, we get min beta I(Z;X) - I(Z;Y). Taking beta = 1/2, multiplying through by 2, and writing I(X;Z) - I(Y Z) = I(X;Z|Y), we find CIB. Unfortunately, I fail to see how this is surprising or different.

A difference only arises when using a variational approximation to IB. The authors compare to the Variational Information Bottleneck (VIB) of Alemi, Fischer, Dillon, and Murphy (arXiv:1612.00410), which requires a classifier, an encoder, and a marginal posterior over the latents. Here, instead of the marginal posterior, they learn a backwards encoder from labels to latents. This difference arises because the IB objective has two terms of opposite sign, and we can group them into positive definite terms in different ways, creating different bounds.

Perhaps this grouping leads to a better variational bound? If so, that's only a point about the variational method employed by Alemi et al., and not a separate objective. As this seems to be the main contribution of the paper, this point needs to be explained more carefully and in more detail. For instance, it seems worth pointing out, in the discrete case, that the marginal posterior |Z| values to estimate, and the backwards encoder has |Z| x |Y| -- suggesting this is a possibly a much harder learning problem. If so, there should be a compelling benefit for using this approximation and not the other one.

In summary, the authors are not really clear about what they are doing and how it relates to IB. Furthermore, the need for this specific choice in IB parameter space is not made clear, nor do the experimental results giving a compelling need. (The experimental results are also not at all clearly presented or explained.) Therefore, I don't think this paper satisfies the quality, clarity, originality, or significance criteria for ICLR.

---

> ### Author Response · Authors · 2018-11-17
> **Response, Part 1**
>
> We appreciate that you read our paper closely, and address your concerns in detail below.
>
>
> Surprise
>
> The surprise of the result relating CEB to IB so simply comes from two things. First, the fact that there is a single value of beta for IB and VIB that achieves the MNI-optimal information, so long as the model, optimizer, etc, is capable of capturing that amount of information. This is surprising because the analysis in Alemi et al. (2018) naively applied to the VIB case would assume that sweeping beta would be necessary to find the optimal information even if you knew a priori what that amount of information was. We discuss this point in the appendix to some extent, but will clarify that discussion in our revisions. Note that a similar and more directly-applicable Pareto-optimal frontier is described in Strauss and Schwab (2017), and that work also does not point out that beta = 1/2 would result in a learned representation where I(X;Z) = I(Y;Z) = I(X;Y).
>
> Second is a point that we decided not to make in the version we submitted, but that we can add in revision. Tishby et al. say two things in many of the IB papers quite clearly.
>
> In Tishby et al. (2015), the authors state:
> “The information bottleneck (IB) method was introduced as an information theoretic principle for extracting relevant information that an input random variable X ∈ X contains about an output random variable Y ∈ Y. Given their joint distribution p(X, Y), the relevant information is defined as the mutual information I(X ; Y), where we assume statistical dependence between X and Y. In this case, Y implicitly determines the relevant and irrelevant features in X. An optimal representation of X would capture the relevant features, and compress X by dismissing the irrelevant parts which do not contribute to the prediction of Y.”
>
> But in Tishby et al. (2000), the authors also state:
> “...there is a tradeoff between compressing the representation and preserving meaningful information, and there is no single right solution for the tradeoff.”
>
> In other words, Tishby et al. recognize that the information measured by I(X;Y) corresponds to the optimal representation, but they do not quite know how to find it. Instead, the IB approach relies on sweeping beta and cross validation.
>
> Thus, the surprise we are describing isn’t the trivial arithmetic relating CEB to IB once you have seen both objectives. Instead, the surprise is that you _can_ learn a representation with the optimal amount of information defined by the observed data, I(X;Y) without having to know I(X;Y) ahead of time. This shows that sweeping beta is unnecessary if you believe (as Tishby appears to believe, and as we certainly believe) that I(X;Y) is the correct amount of information to retain in your representation. The information bottleneck was presented 19 years ago, and so far as we know, no-one has previously proposed a way to learn a representation that doesn’t require you to guess what I(X;Y) might be.
>
>
> Why MNI?
>
> In this work we do not attempt to give a formal proof that CEB representations learn the optimal information about the observed data (and certainly the variational form of the objective will prevent that from happening in general cases). However, the MNI is motivated by the following simple observations: If I(X;Z) < I(X;Y), then we have thrown out relevant information in X for predicting Y. If I(X;Z) > I(X;Y), then we are including information in X that is not useful for predicting Y. Thus targeting I(X;Z) = I(X;Y) is the "correct" amount of information, which is one of the equalities required in order to satisfy the MNI criterion.
>
>
> Relation between general forms of CEB and IB
>
> Yes, CEB is a special case of IB with beta = 1/2. However, we are the first to show that the IB Lagrangian with beta = 1/2 targets the MNI. This is a significant result that was not previously highlighted in any of the literature on IB.

---

> > ### Author Response · Authors · 2018-11-17
> > **Reponse, Part 2**
> >
> > Triviality
> >
> > It is true that beta I(X;Z) - I(Y;Z) is two terms of opposite sign, but it does not immediately follow that you will find the cancelation of H(Z) that we show for CEB unless you start from the observation that you can design a representation learning objective that directly optimizes for covering the information in I(X;Y). To put it another way, of course we can break up IB as follows:
> >
> > beta I(X;Z) - I(Y;Z) = beta / 2 * (H(Z) - H(Z|X) + H(X) - H(X|Z)) - 1 / 2 * (H(Z) - H(Z|Y) + H(Y) - H(Y|Z))
> >
> > There are a number of other obvious ways to split the IB objective as well. But without the key insight about the conditional information term in CEB, and without the guidance of a criterion like the MNI, you have no principle with which to decide to keep or remove any of those entropies or conditional entropies, and since there is a beta associated with one of the H(Z) terms and not the other, you might conclude that you cannot cancel them. As written (in the form that uses both expansions of the mutual information for both terms), the only assignment of beta that allows the H(Z) terms to cancel is beta=1, so you would need to guess that for the I(X;Z) term, you should not use both expansions of I(X;Z), you should only use the H(Z) - H(Z|X) expansion, and then you would have to guess that beta = 1/2 is not just an arbitrary choice of beta, and thus that the cancelation of H(Z) that value permits is worth pursuing. This shows that the CEB objective (in our opinion) is not an obvious consequence of IB, even though once you know both objectives, the relationship between the two is obvious. We hope that the reviewers will keep this perspective in mind while evaluating this work.
> >
> > A similar complaint might be made by observing that b(z|y) is just as valid a variational approximation to p(z) as m(z) is, and so it’s reasonable to learn that as an alternative to VIB. However, you would need to realize that using a “marginal” that depends on y does not give a variational upper bound on I(X;Z). The fact that doing so is part of a bound on I(X;Z|Y) is obvious in retrospect, but we are unaware of anyone pointing this out previously. So again, it is important to not view CEB as merely a particular parameterization of IB, or as VIB with a different variational approximation. These trivializations of the work ignore the important reasoning that allowed us to arrive at such a simple solution to a 19 year old problem.

---

> > > ### Author Response · Authors · 2018-11-17
> > > **Response, Part 3**
> > >
> > > [Apologies for breaking up the response like this. OpenReview gives an unhelpful error message when we try to post the full response, so we are effectively having to perform binary search to determine what part of our text is breaking the site.]
> > >
> > > Ease of optimization
> > >
> > > Your question about optimization in the discrete case is interesting. In IB, the proposed optimization algorithm is the Blahut-Arimoto (BA) algorithm, which converges for a given finite dataset, but adding new data requires retraining. In contrast, VIB is an amortized algorithm, which means that handling new data is trivial, but the results are approximate. For CEB, we chose to focus on the amortized variational approach, but of course we could additionally follow the original IB work and derive similar self-consistent equations for the BA algorithm (for example, trivially by setting beta = 1/2 and using the derivation from IB). In that setting, you are correct that discrete X, Y, and Z would result in a larger search space with CEB than with IB. However, that search would only need to be performed once for CEB, whereas sweeping beta and doing cross validation can reasonably be done until the practitioner runs out of patience, and at the end of that process, the practitioner would still not know if they had learned a representation that covered the information in I(X;Y), unless they already knew that beta=1/2 would give them that result. They wouldn’t know that without this paper or a rediscovery of its results.
> > >
> > > We don’t state this in this paper, but it is clear to us that most of the time, continuous representations are preferable to discrete representations, as they are much easier to work with (and much easier to train via gradient descent). Of course, there are a number of papers in the literature that propose learning discrete representations, or mixed continuous and discrete representations, including InfoGAN (Chen et al., 2016) and Hu et al., 2017. Often, the discrete representation is used to set a hard upper-bound on the amount of information being learned. Since the CEB objective is consistent with the MNI criterion, the learned representation does not need to have such structural constraints placed on it. The practitioner may choose a discrete representation during model specification, but they are not required to do so by the objective. If continuous representations are more convenient, they may use them.
> > >
> > > Finally, our amortized variational training is no slower than VIB -- we train all of the models in the paper for essentially the same number of steps using the same dynamic learning rate schedule to get to convergence. Thus, in the case of continuous representations and amortized inference, a single training of CEB is just as efficient as a single training of VIB, and additionally CEB has no objective function hyperparameters to tune.
> > >
> > >
> > > Variational Tightness
> > >
> > > We are still working out theory to determine how relatively tight the variational approximations are between VIB and variational CEB. We will add any such results in revision if we have them in time. However, informally, when training continuous distributions, it is often easier to train conditional distributions than marginal distributions. Empirically this appears to be the case -- the marginals we train for VIB are mixtures of 500 gaussians, whereas the CEB model’s backward encoder is simply 10 multivariate normal distributions, one for each class, yet the CEB model matches or outperforms the VIB models on all of the tasks. Subsequent experiments with CEB using 10 mixtures of 10 multivariate normals each have proven to be even more effective while still being a less expressive distribution than the 500 component mixture.
> > >
> > >
> > > Experimental Clarity
> > >
> > > We will revise our description of the experiments and add further detail to the appendices that we left out, including more complete descriptions of the modeling and architectural choices, as well as pseudocode for the training algorithm.
> > >
> > >
> > >
> > > [1] Chen, Xi, et al. "Infogan: Interpretable representation learning by information maximizing generative adversarial nets." Advances in neural information processing systems. 2016.
> > > [2] Strouse, D. J., and David J. Schwab. "The deterministic information bottleneck." Neural computation 29.6 (2017): 1611-1630.
> > > [3] Tishby, Naftali, and Noga Zaslavsky. "Deep learning and the information bottleneck principle." Information Theory Workshop (ITW), 2015 IEEE. IEEE, 2015.
> > > [4] Tishby, Naftali, Fernando C. Pereira, and William Bialek. "The information bottleneck method." arXiv preprint physics/0004057 (2000).

---

### Official Review · AnonReviewer4 · 2018-11-15
**A new information bottleneck method is proposed, but major reservations arise**

**Rating:** 6
**Confidence:** 3

**Review:**

[UPDATE]

I find the revised version of the paper much clearer and streamlined than the originally submitted one, and am mostly content with the authors reply to my comments. However, I still think the the work would highly benefit from a non-heuristic justification of its approach and some theoretic guarantees on the performance of the proposed framework (especially, in which regimes it is beneficial and when it is not). Also, I still find the presentation of experimental results too convoluted to give a clear and comprehensive picture of how this methods compares to the competition, when is it better, when is it worse, do the observations/claim generalize to other task, and which are the right competing methods to be considering. I think the paper can still be improved on this aspect as well.

As I find the idea (once it was clarified) generally interesting, I will raise my score to 6.

------------------------------------------------------------------------------------------------------------------------------------------------------------------------

The paper proposes an objective function for learning representations, termed the conditional entropy bottleneck (CEB). Variational bounds on the objective function are derived and used to train classifiers according to the CEB and compare the results to those attained by competing methods. Robustness and adversarial examples detection of CEB are emphasized.

My major comments are as follows:

1) The authors base their 'information-theoretic' reasoning on the set-theoretic structure of Shannon’s information measures. It is noteworthy that when dealing with more than 2 random variables, e.g., when going from the twofold I(X;Y) to the threefold I(X;Y;Z), this theory has major issues. In particular, there are simple (and natural) examples for which I(X;Y;Z) is negative. The paper presents an information-theoretic heuristic/intuitive explanation for their CEB construction based on this framework. No proofs backing up any of the claims of performance/robustness in the paper are given. Unfortunately, with such counter-intuitive issues of the underlying theory, a heurisitc explanation that motivates the proposed construction is not convincing. Simulations are presented to justify the construction but whether the claimed properties hold for a wide variety of setups remain unclear.

2) Appendix A is referred to early on for explaining the minimal necessary information (MNI), but it is very unclear. What is the claim of this Appendix? Is there a claim? It's just seems like a convoluted and long explanation of mutual information. Even more so, this explanation is inaccurate. For instance, the authors refer to the mutual information as a 'minimal sufficient statistic' but it is not. For a pair of random variables (X,Y), a sufficient statistic, say, for X given Y is a function f of Y such X-f(Y)-Y forms a Markov chain. Specifically, f(Y) is another random variable. The mutual information I(X;Y) is just a number. I have multiple guesses on what the authors' meaning could be here, but was unable to figure it out from the text. One option, which is a pretty standard way to define sufficient statistic though mutual information is as a function f such that I(X;Y|f(Y))=0. Such an f is a sufficient statistic since the zero mutual information term is equivalent to the Markov chain X-f(Y)-Y from before. Is that what the authors mean..?

3) The Z_X variable introduced in Section 3 in inspired by the IB framework (footnote 2). If I understand correctly, this means that in many applications, Z_X is specified by a classifier of X wrt the label Y. My question is whether for a fixed set of system parameters, Z_X is a deterministic function of X? If this Z_X play the role of the sufficient statistics I've referred to in my previous comment, then it should be just a function of X.

However, if Z_X=f(X) for a deterministic function f, then the CEB from Equation (3) is vacuous for many interesting cases of (X,Y). For instance, if X is a continuous random variable and Z_X=f(X) is continuous as well, then
I(X;Z_X|Y)=h(Z_X|Y)-h(Z_X|X,Y)
where h is the differential entropy and the subtracted terms equals -\infty by definition (see Section 8.3 of (Cover & Thomas, 2006). Consequently, the mutual information and the CEB objective are infinite. If Z_X=f(X) is a mixed random variable (e.g., can be obtain from a ReLU neural network), then the same happens. Other cases of interest, such as discrete X and f being an injective mapping of the set of X values, are also problematic. For details of such problem associated with IB type terms see:

[1] R. A. Amjad and B. C. Geiger 'Learning Representations for Neural Network-Based Classification Using the Information Bottleneck Principle', 2018 (https://arxiv.org/abs/1802.09766).

Can the authors account for that?

4) The other two reviews addressed the missing accounts for past literature. I agree on this point and will keep track of the authors' responses. I will not comment on that again.

Beyond these specific issue, they text is very wordy and confusing at times. If some mathematical justification/modeling was employed the proposed framework might have been easier to accept. The long heuristic explanations employed at the moment do not suffice for this reviewer. Unless the authors are able to provide clarification of all the above points and properly place their work in relation to past literature I cannot recommend acceptance.

---

> ### Author Response · Authors · 2018-11-17
> **Response**
>
> Thank you for taking the time to write a detailed review. We will address your concerns in turn.
>
> Negativity of I(X;Y;Z)
>
> You are correct that I(X;Y;Z) can be negative in general. However, it cannot be negative in the representation learning setting that we are describing here -- the Markov chain Z <- X -> Y does not permit it.
>
> The triplet information, I(X;Y;Z) may be defined as follows:
>
> I(X;Y;Z) = I(X;Z) - I(X;Z|Y)
>
> But we already know that I(X;Z|Y) = I(X;Z) - I(Y;Z) (Equation 2) due to our Markov chain Z <- X <-> Y, so in our case we have:
>
> I(X;Y;Z) = I(X;Z) - I(X;Z) + I(Y;Z) = I(Y;Z)
>
> which we also know is non-negative, completing our proof.
>
>
> Minimum Necessary Information
>
> Our upcoming revisions clarify the relationship between MNI and minimal sufficient statistics and update the discussion of the MNI with the following:
>
> Why MNI?
>
> In this work we do not attempt to give a formal proof that CEB representations learn the optimal information about the observed data (and certainly the variational form of the objective will prevent that from happening in general cases). However, the MNI is motivated by the following simple observations: If I(X; Z) < I(X; Y), then we have thrown out relevant information in X for predicting Y. If I(X; Z) > I(X; Y), then we are including information in X that is not useful for predicting Y. Thus targeting I(X; Z) = I(X; Y) is the "correct" amount of information, which is one of the equalities required in order to satisfy the MNI criterion.
>
>
> Representations and Finiteness of Mutual Information
>
> Z_X is not a deterministic function of X in either IB or CEB, it is a stochastic representation of X given by an encoder e(z_X|x). Using a stochastic encoder means that the stated concerns of infinite entropy are not applicable in either objective -- the conditions for infinite mutual information given in Amjad and Geiger (2018) do not apply. In practice, (using continuous representations), we did not encounter mutual information terms that diverged to infinity, although certainly it is possible to make modeling and data choices that make it more or less likely that there will be numerical instabilities. This is not a flaw specific to CEB or VIB, however, and we found numerical instability to be almost non-existent across a wide variety of modeling and architectural choices for both variational objectives.
>
>
> [1] R. A. Amjad and B. C. Geiger 'Learning Representations for Neural Network-Based Classification Using the Information Bottleneck Principle', 2018

---

> > ### Comment · AnonReviewer4 · 2018-11-17
> > **Accept the answer to the first two comments (though still lack rigor in the general approach) and a question regarding the third**
> >
> > The authors responses to the first two comments are satisfactory. These responses clarified much that was hard to understand from the actual text. This suggest a revision on the explanations in the manuscript is due. The idea as elaborated in the response is appealing, but the text was too broad and fancy to pin down this concrete concept (that preserving I(X;Y) information in Z_X about X and Y is a reasonable thing to do and why).
> >
> > On the flip side, it would be *very* nice to see some proofs justifying that performance (in an appropriate sense) degrades when I(X; Z) < I(X; Y) or (X; Z) > I(X; Y) happens. This would *significantly* strengthen the work.
> >
> > By the way, there is no need to call Eqn. (2) a 'well-known equality' and to cite Thomas & Cover. It is just I(X;Z|Y)=I(X,Y;Z)-I(Y;Z)=I(X;Z)-I(Y;Z), where the first step in mutual information chain rule, and the second step is the Markob chain (essentially that conditional distribution satisfies P_{Z|X,Y=y}=P_{Z|X} for all y, and some algebra). Just saying that it follows by the above would suffice. Also, the notation Z<-X<->Y is redundant and confusing. It is always the case that if Z<-X<-Y forms a Markov chain in that order then Z->X->Y is another valid ordered chain. Consequently, we can just write Z<->X<->Y, or even better: Z-X-Y.
> >
> > All in all, I look forward to the revision, hoping it will reflect the clarity of the response.
> >
> > Regarding the last point, if Z_X is implemented by a DNN, when the parameters of the system are fixed, what makes it (or the encoder e(z_X|x) for that matter) stochastic? Can the authors specify their assumptions on Z_X (or the family of models) that preclude it from being deterministic, for fixed parameters? Where does this construction endow Z_X with a mechanism for shedding information about X -- is there quantization? noise?

---

> > > ### Author Response · Authors · 2018-11-17
> > > **Response to question on the third point**
> > >
> > > Thank you for your rapid reply, and for continuing to engage with us as we improve our presentation. We look forward to your feedback once we post our revisions.
> > >
> > > The simplest way of parameterizing a distribution with a DNN involves using the reparameterization trick from Kingma and Welling (2013), which is most easily understood as including a sample from a standardized version of the desired distribution as an additional input to the network that gets incorporated at the last layer of the encoder to give a valid sample from the desired distribution. More recently, Figurnov et al. (2018) showed how to allow many more distributions to be reparameterized. Finally, it is also possible to use score function estimators to take unbiased but high-variance gradients through non-reparameterizable or discrete distributions.
> > >
> > > Given these options, there are very few limitations on how to set up your desired Z_X distribution, although some options will be easier to train than others. In the VAE and VIB literature, using a multivariate normal distribution is quite common for the encoder, as the expressivity of the underlying DNN typically can adjust the input space sufficiently that the resulting encoder distributions can give useful latent samples for the desired task. Our experiments with these types of models typically start with fully covariant multivariate normal distributions, but we have also used multivariate beta distributions and other more esoteric choices for the encoder.
> > >
> > > (Please let us know if we have missed the thrust of your questions.)
> > >
> > > [1] Kingma, Diederik P., and Max Welling. "Auto-encoding variational bayes." arXiv preprint arXiv:1312.6114 (2013).
> > > [2] Figurnov, Michael, Shakir Mohamed, and Andriy Mnih. "Implicit Reparameterization Gradients." arXiv preprint arXiv:1805.08498 (2018).

---

### Author Response · Authors · 2018-11-17
**General Response**

We would like to thank the reviewers for their efforts and consideration in reviewing our work. Most of the concerns about the work lie with how easy it is to comprehend the motivation of the minimum necessary information criterion and the derivation of the CEB objective, which we will clarify in our revisions. Individual concerns are addressed in the corresponding threads. Here, we would like to make two high-level points, one about the core purpose of the paper, and the other about the empirical results.

All three of the remaining reviews (since one was deleted) seem to have not understood that we are providing a way to learn a representation Z of observed joint data, X and Y, that maintains only the amount of information I(X;Y) about both X and Y, and does so without any hyperparameter tuning. (Only the deleted review that scored the work an 8 gave a summarization that indicated that the reviewer had understood this point.) The premise of MNI, and what makes it more than just “a convoluted and long explanation of mutual information” is that it is defining the amount of information the optimal representation, Z, should have about each of the two observed variables individually and collectively. We are somewhat surprised that this was unclear, as that goal is given as the first formula of Section 3: I(X;Y;Z) = I(X;Z) = I(Y;Z) = I(X;Y), and explained diagrammatically in Figure 1 and the text that references Figure 1. This differs from the statement of the Information Bottleneck principle (Tishby et al., 2000), although Tishby et al. do acknowledge in subsequent work that I(X;Y) is the optimal amount of information a learned representation should have (Tishby et al., 2015). Even with their recent admission that I(X;Y) is the optimal amount of information, they do not know how to set beta to find that amount of information. Thus, our surprising result is that this amount of information, I(X;Y), can be directly targeted for a given representation Z without knowing in advance the value of I(X;Y), and without any hyperparameter tuning.

Little review attention has been given to the empirical results of the work (one review finds the experiments difficult to understand, two reviews request additional experiments but don’t mention that the experiments we did are problematic or unconvincing, the fourth review does not comment on the experiments at all). Our empirical results show a strong advantage to using the CEB objective in an apples-to-apples comparison on four major outstanding issues in the field of machine learning, all of which can be characterized as problems of generalization.

In particular, we consider our results with adversarial robustness and detection to be compelling -- the whitebox attacks we experimented with amount to an essentially unlimited whitebox adversary, and we are unaware of any work showing real robustness to even the basic version of the adversary, much less the adversary that additionally directly targets the detection mechanism. Recent work (Carlini et al., 2018) urges researchers in the space to try to attack their own mechanism -- our experiments do exactly that, and the remarkable result is that the model is even more robust to those attacks (while less able to detect them). Certainly we have not encountered any work that confers adversarial robustness merely by changing the objective function used to train an otherwise identical inference network on an unmodified dataset. That we are able to achieve those results and the others described in the paper while maintaining performance parity on the core classification task underscores why we expect this paper to have broad interest. A simple change to the objective function used to train the model (no harder to implement than a VAE or VIB model), with no performance lost relative to standard maximum likelihood techniques, while gaining much in terms of generalization performance, all without additional hyperparameters to tune seems to us like something that many researchers and practitioners will want to explore. Given that our objective is motivated from a representation learning perspective, it is difficult to imagine a more appropriate venue for the work than ICLR.

We hope that the reviewers will take the time to consider these strong empirical results while reviewing our revisions.

[1] Athalye, Anish, Nicholas Carlini, and David Wagner. "Obfuscated gradients give a false sense of security: Circumventing defenses to adversarial examples." ICML (2018).
[2] Tishby, Naftali, Fernando C. Pereira, and William Bialek. "The information bottleneck method." arXiv preprint physics/0004057 (2000).
[3] Tishby, Naftali, and Noga Zaslavsky. "Deep learning and the information bottleneck principle." Information Theory Workshop (ITW), 2015 IEEE. IEEE, 2015.

---

> ### Author Response · Authors · 2018-11-17
> **Response to Deleted Review**
>
> We are disappointed that one of our reviews was deleted, as it also gave useful feedback on our work (in addition to giving our work a very positive rating, which we of course appreciated). We will not copy the original review here, but we would like to give our response regardless.
>
> Thank you for your careful summary of our work. We have continued experimenting since submission, including training CEB models on CIFAR10, where we have achieved almost 95% test accuracy with wide resnets of varying depths and basic data augmentation. We would be happy to add these results in revision if reviewer consensus is that these are critical to the acceptance of the paper.
>
> We would also like to point out (and we will clarify in revision) that any distributional family may be used for the encoder -- reparameterizable distributions are convenient, but it is also possible to use the score function trick to get a high-variance estimate of the gradient for distributions that have no explicit or implicit reparameterization. In general, a good choice for b(z|y) is a mixture of whatever the encoder distribution is. The core point is that these are modeling choices that need to be made by the practitioner, and they depend very much on the dataset (as you suggest). In this work, we chose normal distributions because they are easy to work with and will be the common choice for many problems, particularly when parameterized with neural networks, but that choice is incidental rather than fundamental.
>
> You are correct that we did no explicit regularization on the deterministic model. We think it is likely that a regularized model would perform somewhat better at adversarial robustness, depending on the choice of regularizer. However, we don’t think it’s a bold claim to say that none of the standard regularization techniques provide meaningful robustness to the CW attack (which typically achieves 100% attack success rate on models that are not specifically designed to thwart it). It isn’t clear to us how much standard regularization techniques impact the other tasks on this particular dataset. Perhaps classification performance would have improved slightly? Our results are very much in-line with the Fashion MNIST leaderboard (https://github.com/zalandoresearch/fashion-mnist) for small networks -- in fact, 93% test accuracy or greater is only achieved by much larger or more sophisticated networks (VGG16, Capsule Networks, etc), apart from one simple convnet result -- so we think our results are likely to be representative. Finally, we would like to note that, even though there is no explicit regularization, all of the networks use the same size latent space, which means that the deterministic network also has a 4D bottleneck layer. We imagine that was helpful in minimizing overfitting for the deterministic network.

---

> > ### Comment · Area_Chair1 · 2018-11-17
> > **Review was deleted due to a conflict of interest**
> >
> > The deleted review was deleted because the reviewer discovered a conflict of interest.  It's unfortunate that this only came to light after the beginning of the discussion period.

---

> ### Public Comment · (anonymous) · 2018-11-24
> **related work**
>
> Before Alemi et al. (2017), there was Achille et al. https://arxiv.org/pdf/1611.01353 on the topic. Also relevant to this discussion is https://arxiv.org/pdf/1706.01350.

---

> > ### Author Response · Authors · 2018-11-27
> > **References**
> >
> > Thank you for reminding us to cite InfoDropout. It is included in our latest revision. We agree that the Emergence of Invariance paper is related, although proper treatment will require some care, so we did not manage to fit it into the current set of revisions.

---

### Author Response · Authors · 2018-11-27
**Revision Submitted**

We have finished revisions based on reviewer feedback. In addition to addressing reviewer concerns, we have added simple geometric analyses of both CEB and IB which we think substantially clarify our contribution. We have also formalized the Minimum Necessary Information criterion. We look forward to further feedback and discussion from reviewers.

---

### Meta-Review · Area_Chair1 · 2018-12-13
**Somewhat controversial, but interesting new criterion for representation learning**

**Confidence:** 4
**Recommendation:** Reject

**Metareview:**

This paper proposes a criterion for representation learning, minimum necessary information, which states that for a task defined by some joint probability distribution P(X,Y) and the goal of (for example) predicting Y from X, a learned representation of X, denoted Z, should satisfy the equality I(X;Y) = I(X;Z) = I(Y;Z). The authors then propose an objective function, the conditional entropy bottleneck (CEB), to ensure that a learned representation satisfies the minimum necessary information criterion, and a variational approximation to the conditional entropy bottleneck that can be parameterized using deep networks and optimized with standard methods such as stochastic gradient descent. The authors also relate the conditional entropy bottleneck to the information bottleneck Lagrangian proposed by Tishby, showing that the CEB corresponds to the information bottleneck with β = 0.5. An important contribution of this work is that it gives a theoretical justification for selecting a specific value of β rather than testing multiple values. Experiments on Fashion-MNIST show that, in comparison to a deterministic classifier and to variational information bottleneck models with β in {0.01, 0.1, 0.5}, the CEB model achieves good accuracy and calibration, is competitive at detecting out-of-distribution inputs, and is more resistant to white-box adversarial attacks. Another experiment demonstrates that a model trained with the CEB criterion is *unable* to memorize a randomly labeled version of Fashion-MNIST. There was a strong difference of opinion between the reviewers on this paper. One reviewer (R1) dismissed the work as trivial. The authors rebutted this claim in their response and revision, and R1 failed to participate in the discussion, so the AC strongly discounted this review. The other two reviewers had some concerns about the paper, most of which were addressed by the revision. But, crucially, some concerns still remain. R4 would like more theoretical rigor in the paper, while R2 would like a direct comparison against MINE and CPC. In the end, the AC thinks that this paper needs just a bit more work to address these concerns. The authors are encouraged to revise this work and submit it to another machine learning venue.